# The Value of Citizen Science for Flood Risk Reduction: Cost-benefit Analysis of a Citizen Observatory in the Brenta-Bacchiglione Catchment

Michele Ferri[1], Uta Wehn[2], Linda See[3], Martina Monego[1], Steffen Fritz[3]

[1]Alto-Adriatico Water Authority/Autorità di bacino distrettuale delle Alpi orientali (AAWA), Cannaregio 4314, 30121 Venice, Italy
[2]IHE Delft Institute for Water Education, Westvest 7, 2611 AX Delft, The Netherlands
[3]International Institute for Applied Systems Analysis (IIASA), Schlossplatz 1, 2361 Laxenburg, Austria

*Correspondence to*: Michele Ferri (michele.ferri@distrettoalpiorientali.it)

**Abstract.** Citizen observatories are a relatively recent form of citizen science. As part of the flood risk management strategy of the Brenta-Bacchiglione catchment, a citizen observatory for flood risk management has been proposed and is currently being implemented. Citizens are involved through monitoring water levels and obstructions and providing other relevant information through mobile apps, where the data are assimilated with other sensor data in a hydrological-hydraulic model used in early warning. A cost benefit analysis of the citizen observatory was undertaken to demonstrate the value of this approach in monetary terms. Although not yet fully operational, the citizen observatory is assumed to decrease the social vulnerability of the flood risk. By calculating the hazard, exposure and vulnerability of three flood scenarios (required for flood risk management planning by the EU Directive on Flood Risk Management) with and without the proposed citizen observatory, it is possible to evaluate the benefits in terms of the average annual avoided damage costs. Although currently a hypothetical exercise, the results showed a reduction in avoided damage of 45% compared to a business as usual scenario. Thus, linking citizen science and citizen observatories with hydrological modelling to raise awareness of flood hazards and to facilitate two-way communication between citizens and local authorities has great potential in reducing future flood risk in the Brenta-Bacchiglione catchment. Moreover, such approaches are easily transferable to other catchments.

## 1 Introduction

In 2018, flooding affected the highest number of people of any natural disaster globally and caused major damage worldwide (CRED, 2019). With climate change, the frequency and magnitude of extreme events will increase, leading to a higher risk of flooding (Schiermeier, 2011). This risk will be further exacerbated by future economic and population growth (Tanoue et al., 2016). Thus, managing flood risk is critical for reducing future negative impacts. Flood risk assessments are undertaken by the insurance industry for determining properties at high risk (Hsu et al., 2011), but they are also a national requirement in the European Union as set out in the EU Flood Risk Management Directive, which requires that flood risk management plans are produced for each river basin (EU, 2007; Müller, 2013). The assessment of flood risk involves quantifying three main drivers (National Research Council, 2015): (a) flood hazard, which is the probability that a flood of a certain magnitude will occur in a certain period of time in a given area; (b) exposure, which is the economic value of the human lives and assets affected by the flood hazard; and (c) vulnerability, which is the degree to which different elements (i.e., people, buildings, infrastructure, economic activities, etc.) will suffer damage associated with the flood hazard. In addition, flood risk can be mitigated through hard engineering strategies such as implementation of structural flood protection schemes, soft engineering approaches comprising more natural methods of flood management (Levy and Hall, 2005), and community-based flood risk management (Smith et al., 2017). As part of requirements in the EU Flood Risk Management Directive, any mitigation actions must be accompanied by a cost-benefit analysis.

Flood hazard is generally determined through hydrological and hydraulic modelling. Hence accurate predictions are critical for effective flood risk management, particularly in densely populated urban areas (Mazzoleni et al., 2017). The input data required for modelling are often incomplete in terms of resolution and density (Lanfranchi et al., 2014), which translates into variable accuracy in flood predictions (Werner et al., 2005). New sources of data are becoming available to support flood risk management. For example, the rise of citizen science and crowdsourcing (Howe, 2006; Sheldon and Ashcroft, 2016), accelerated by the rapid diffusion of information and communication technologies, is providing additional, complementary sources of data for hydrological monitoring (Njue et al., 2019). Citizen science refers to the involvement of the public in any step of the scientific method (Shirk et al., 2012). However, one of the most common forms of participation is in data collection (Njue et al., 2019). Citizen observatories (CO) are a particular form of citizen science in so far as they constitute the means not just for new knowledge creation but also for its application, which is why they are typically set up with linkages to specific policy domains (Wehn et al., 2019). COs must, therefore, include a public authority (e.g., a local, regional or national body) to enable two-way communication between citizens and the authorities to create a new source of high quality, authoritative data for decision making and for the benefit of society. Moreover, COs involve citizens in environmental observations over an extended period of time of typically months and years (rather than one-off exercises such as data collection 'Blitzes'), and hence contribute to improving the temporal resolution of the data, using dedicated apps, easy-to-use physical sensors and other monitoring technologies linked to a dedicated platform (Liu et al., 2014; Mazumdar et al., 2016). COs are increasingly being used in hydrology/water sciences and management and in various stages of the flood risk management cycle, as reviewed and reported by Assumpção (2018), Wehn and Evers (2015) and Wehn et al. (2015). Specifically, Wehn et al. (2015) found that the characteristic links of COs to authorities and policy do not automatically translate into higher levels of participation in flood risk management, nor that communication between stakeholders improves; rather, changes towards fundamentally more involved citizen roles with higher impact in flood risk management can take years to evolve.

The promising potential of the contribution of COs to improved flood risk management is paralleled by limited evidence of their actual impacts and added value. Efforts are ongoing such as the consolidation of evaluation methods and empirical evidence by the H2020 project WeObserve[1] Community of Practice on the value and impact of citizen science and COs, and the development and application of methods for measuring the impacts of citizen science by the H2020 project MICS[2]. To date, the societal and science-related impacts have received the most attention, while the focus on economic impacts, costs and benefits has been both more limited and more recent (Wehn et al., 2020a). The studies that do focus on economic impacts related to citizen science (rather than citizen observatories) propose to consider the time invested by researchers in engaging and training citizens (Thornhill et al., 2016); to relate cost and participant performance for hydrometric observations in order to estimate the cost per observation (Davids et al., 2019); to estimate the costs as data-related costs, staff costs and other costs; and the benefits in terms of scientific benefits, public engagement benefits and the benefits of strengthened capacity of participants (Blaney et al., 2016); and to compare citizen science data and in-situ data (Goldstein et al., 2014; Hadj-Hammou et al., 2017). Wehn et al. (2020b) assessed the value of COs from a data perspective and a cost perspective, respectively, to qualify the degree of complementarity that the data collected by citizens offer to in-situ networks and to quantify the relation between the investments required to set up a CO and the actual amount of data collected. Based on a comparison of four COs, they suggest that setting up a CO for the sole purpose of data collection appears to be an expensive undertaking (for the public sector organization(s) benefitting from the respective CO) since, depending on the process of (co)designing the CO, it may not necessarily complement the existing in-situ monitoring network (with the likely exception of infrastructure-weak areas in developing countries).

---

[1] https://www.weobserve.eu/
[2] https://mics.tools/

Overall, there is a lack of available, appropriate and peer-reviewed evaluation methods and of evidence of the added value of COs, which is holding back the uptake and adoption of COs by policy makers and practitioners. In this paper, we take a different approach to previous studies by using a more conventional cost-benefit analysis framework to assess the implementation of a CO on flood risk management in the Brenta-Bacchiglione catchment in northern Italy. The purpose of a cost-benefit analysis is to compare the effectiveness of different alternative actions, where these actions can be public policies, projects or regulations that can be used to solve a specific problem. We treat the CO in the same way as any other flood mitigation action for which a cost-benefit analysis would be undertaken in this catchment. Although the CO is still being implemented, the assumptions for the cost-benefit analysis are based on primary empirical evidence from a CO pilot that was undertaken by the WeSenseIt project in the town of Vicenza, Italy, described in more detail in section 2.1 and now extended to the wider catchment (sections 2.2 and 2.3). In section 3 we present the flood risk and cost benefit methodology followed by the results in section 4. Conclusions, limitations of the methodology and case-specific insights are provided in section 5.

## 2 The Development of a Citizen Observatory for Flood Risk Management

### 2.1 The WeSenseIt Project

Through the WeSenseIt research project (www.wesenseit.eu), funded under the $7^{th}$ framework program (FP7-ENV-2012 n° 308429), a CO for flood risk management was developed with the Alto Adriatic Basin Authority in northern Italy. The objective of this CO was to collect citizen observations from the field, and to obtain a broader and more rapid picture of developments before and during a flood event. The CO involved many stakeholders concerned with the management and use of the water resources, and with water-related hazards in the Bacchiglione River basin. The main actors included the local municipalities, the regional and local civil protection agencies, environment agencies and the irrigation authorities. The Alto Adriatico Water Authority (AAWA) facilitated access to a highly trained group of citizen observers, namely civil protection volunteers, who undertook the observations (i.e., using staff gauges with a QR code to measure the water level and reporting water way obstructions – see **Figure** 1) as part of their volunteer activities. Additional volunteers were also recruited during the project from the Italian Red Cross, the National Alpine Trooper Association, the Italian Army Police and other civil protection groups, with more than 200 volunteers taking part in the CO pilot. Training courses for the volunteers were organized to disseminate and explain the use of a smartphone application and an e-collaboration platform, which were developed as part of the WeSenseIt project. In addition to the low cost sensing equipment, the CO also used data from physical sensors, which are operated by AAWA in collaboration with the Regional Department for Soil Protection, the Environmental Agency, and the Civil Protection Agency including: 3 sonar sensors (river water level), 4 weather stations (wind velocity and direction, precipitation, air temperature and humidity) and 5 soil moisture sensors. The combined visualization of the sensors (including existing sensors from the Venice Environment Agency) was available in the online e-collaboration platform. During the WeSenseIt project, research into the value of crowdsourced data for hydrological modelling was investigated (Mazzoleni et al., 2017, 2018) and found to complement traditional sensor networks.

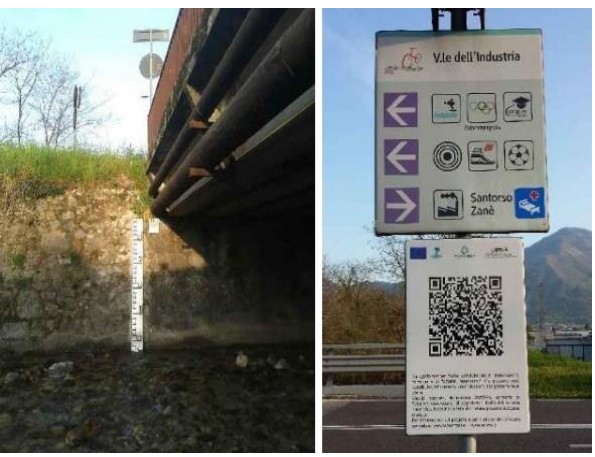

**Figure 1: Photos showing staff gauges and QR codes used in the WeSenseIt project.**

This pilot was later adopted by the European Community as a "good practice" example of the application of Directive 2007/60/EC. After the positive experience in WeSenseIt, funds were made available to develop a CO for flood risk management at the district scale, covering the larger Brenta-Bacchiglione catchment. At this stage, a cost-benefit analysis was undertaken, which is reported in this paper. The next section provides details of the Brenta-Bacchiglione catchment followed by ongoing developments in the CO for flood risk management.

**2.2 The Brenta-Bacchiglione Catchment**

The Brenta-Bacchiglione River catchment falls within the Trento-Alto-Adige and Veneto Regions in Northern Italy and includes the cities of Padua and Vicenza (**Figure** 2). The catchment is surrounded by the Beric hills in the south and the Prealpi in the northwest. In this mountainous area, rapid or flash floods occur regularly and are difficult to predict. Rapid floods generally affect the towns of Torri di Quartesolo, Longare and Montegaldella, although there is also widespread

flooding in the cities of Vicenza and Padua, which includes industrial areas and areas of cultural heritage. For example, in 2010, a major flood affected 130 communities and 20,000 individuals in the Veneto region. The city of Vicenza was one of the most affected municipalities, with 20% of the metropolitan area flooded.

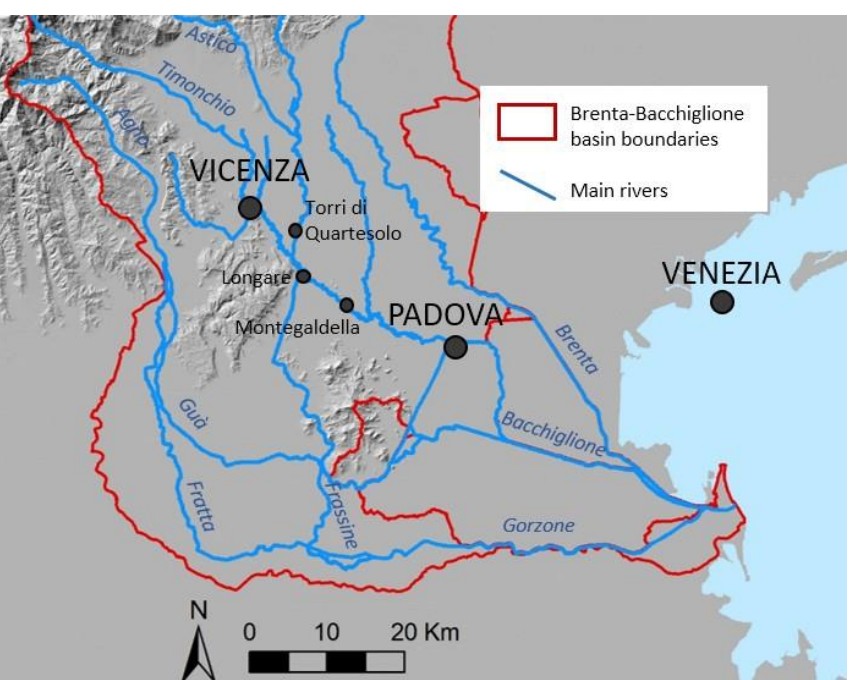

**Figure 2: Location of the Brenta-Bacchiglione catchment and its urban communities.**

**2.3 The Citizen Observatory for Flood Risk Management for the Brenta-Bacchiglione Catchment**

The CO for flood risk management, which is currently being implemented, was included in the prevention measures of the Flood Risk Management Plan (PGRA) for the Brenta-Bacchiglione catchment. The purpose of the CO is to strengthen communication channels before and during flood events in accordance with the EU Flood Directive on Flood Risk Management, to increase the resilience of the local communities and to address residual risk. Building on the WeSenseIt experience, an IT platform to aid decision support during the emergency phases of a flood event is being implemented. This platform will integrate information from the hydrological model, which is equipped with a data assimilation module that integrates the crowdsourced data collected by citizens and trained experts with official sensor data. A mobile app for data collection based on the WeSenseIt project is under development. The platform and mobile technology will guarantee user traceability and facilitate two-way communication between the authorities, the citizens and the operators in the field, thereby significantly increasing the effectiveness of civil protection operations during all phases of an emergency. The fully operational CO will include 64 additional staff gauges equipped with a QR code (58 to measure water level and 6 for snow height), 12 sonar sensors and 8 weather stations.

To engage and maintain the involvement of "expert" CO participants (i.e., civil protection volunteers, technicians belonging to professional associations, members of environmental associations), a set of training courses will be run. The involvement of technicians (formalized in November 2018 through an agreement between the respective associations and AAWA) offers an important opportunity to use the specific knowledge and expertise of these technicians to better understand the dynamics of flood events and to acquire high quality data to feed the models and databases. When an extreme event (i.e., heavy rain) is forecast, AAWA will call upon any available technicians in providing data (with a reimbursement of 75 €/day (including insurance costs) and a minimum activity per day of 3 hours). There are currently 41 technicians involved in the CO, which includes civil/hydraulic/geotechnical engineers, agronomists and forestry graduates. Participants must attend two training sessions followed by a final examination. To give an example of the valuable information that the expert CO participants can provide, AAWA called upon technicians during two heavy rainfall events (November 2019; 5 days). These technicians collected relevant data on the status of the rivers including the vegetation, the water levels, and the status of bridges and levees, collecting 1660 images and completing 700 status reports.

To engage citizens, a different approach is being taken. Within the 120 municipalities currently in high flood risk zones, engagement of schools is currently ongoing, including the development of educational programs for teachers. The aim is to raise student awareness of existing flood risks in their own area, and to help students recognize the value of the CO (and the mobile technology) in protecting their families, e.g., using the app to send and share reports regarding the water level of a river at a section equipped with a hydrometric measuring rod and QR code, the level of the snowpack from a snow gauge equipped with a QR code, the presence of flooded areas including the water height, as well as simplified measurements of hydrological variables such as the amount of rain, weather conditions, etc. using photographs and other smart ways to identify the phenomenon. By providing important information about flooding, this will contribute to everyone's safety. In exchange, citizens can receive flood-related information (e.g., weather and river level forecasts, notifications from the authority concerning the declaration of a state of alert or its cessation, specific communications to citizens present in a specific area of interest/danger in a specific period of time, based on a geolocation function). This two-way communication can help to reduce flood risks. This component of the CO involves 348 primary schools and 340 middle and secondary schools. The three universities in the area will also be involved through conferences and webinars. Communication through the CO website, via social media campaigns, radio broadcasts and regional newspapers will be used to engage and maintain citizen involvement in the CO. This communication plan, which will continue over the next five years, has the ambitious goal of involving 75,000 people in the CO to download the app and contribute observations.

## 3 Methodology

The methodology consists of two steps: (i) mapping of the flood risk (section 3.1); and (ii) quantification of the flood damage costs (section 3.2), which consider the flood risk with and without the implementation of the CO on flood risk management.

### 3.1 Flood risk mapping

Figure 3 provides an overview of the flood risk methodology employed in the paper, which uses input data outlined in section 3.1.1. As mentioned in the introduction, risk is evaluated from three different components. The first is the flood hazard, which is calculated using a hydrological-hydraulic model to generate flood hazard maps and is described in section 3.1.2. The second is exposure, outlined in section 3.1.3, which is calculated for three macro-categories as set out in the EU 2007/60/CE Flood Directive (EU, 2007): the population affected (art.6-5.a); the types of economic activities affected (art.6-5.b); and the environmental and cultural-archaeological assets affected (art.6.5.c).

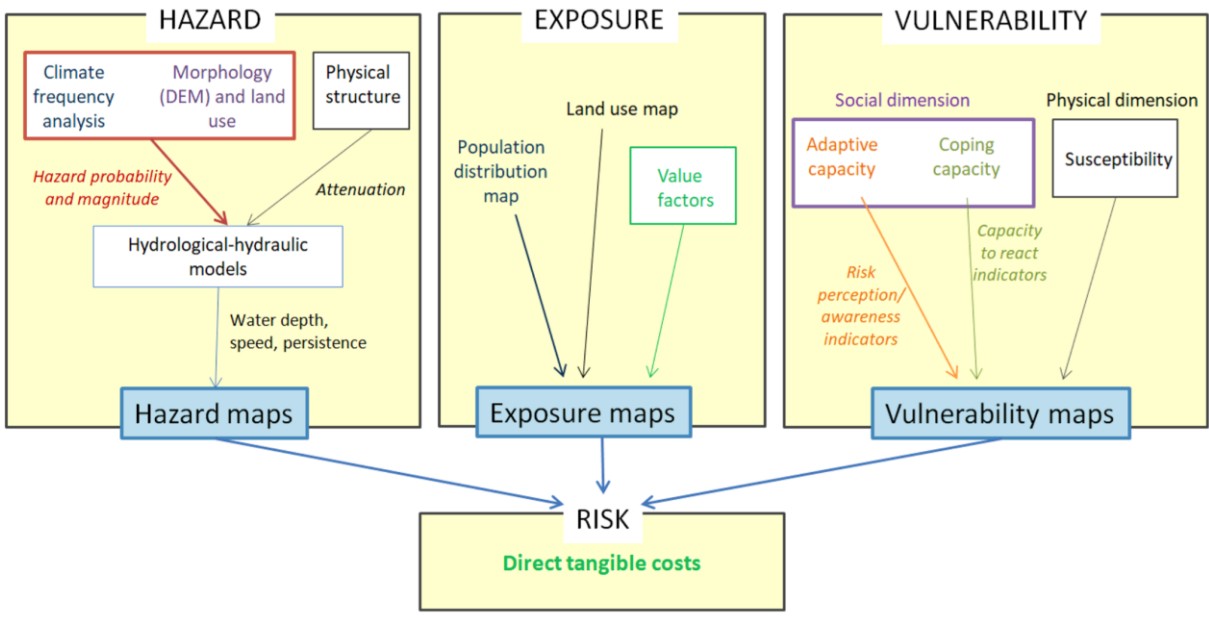

**Figure 3: Flowchart outlining the determination of risk in a flood risk assessment context.**

The final component is vulnerability, which has a physical and social dimension. Physical vulnerability is defined as the susceptibility of an exposed element such as people or buildings to flooding (Balbi et al., 2012) and is calculated using the same three macro-categories as that of exposure, i.e., the population affected, the economic activities affected, and the environmental and cultural-archaeological assets affected. Within the people affected category, we also consider social vulnerability. This refers to the perception or awareness that an adverse event may occur. Some studies have found that if citizens have directly experienced a flood, their perception of flood risk is higher (e.g., Thistlethwaite et al., 2018) although the factors that determine flood risk perception are varied. Moreover, the results from different studies can be ambiguous and/or contradictory (Lechowska, 2018). Social vulnerability can be divided into: (i) adaptive capacity, which is the capacity of an individual, community, society or organization to prepare for and respond to the consequences of a flood event (IPCC, 2012; Torresan et al., 2012); and (ii) coping capacity, which is the ability of an individual, community, society or organization to cope with adverse conditions resulting from a flood event using existing resources (IPCC, 2012; Torresan et al., 2012). The calculation of vulnerability is described in section 3.1.4. Risk is then calculated as the product of hazard, exposure and vulnerability as described in more detail in section 3.1.5, from which the direct tangible costs associated with

the flood risk can be calculated (outlined in section 3.2). The model assumptions and the sources of uncertainty are summarized in Table S1 in the Supplementary Material.

### 3.1.1 Input data

There are several data sets used as inputs to the assessment of flood risk as outlined in Table 1. For the evaluation of flood hazard, the water height, flow velocity and flooded areas are provided by AAWA using the methodology described in the Supplementary Materials. Several data sets are used to evaluate flood exposure and vulnerability, but a key data set is Corine Land Cover 2006 produced by the European Environment Agency (Steemans, 2008). Other data sets used to determine exposure include layers on population, infrastructure and buildings, areas of cultural heritage, protected areas and sources of pollution, where these data sets were obtained from different Italian ministries to complement the Corine Land Cover. Data from OpenStreetMap on infrastructure and buildings were also used.

**Table 1: Input data used to calculate risk.**

| Component of risk | Data | Source |
|---|---|---|
| Flood Hazard (low, medium, high hazard scenarios) | Water height (m) | AAWA; see Supplementary Materials for model details |
| | Water velocity (m/s) | |
| | Flooded area ($km^2$) | |
| Flood Exposure | Population in residential areas | ISTAT, census data, 2001 |
| | Infrastructure and buildings | Corine Land Cover 2006, OpenStreetMap |
| | Types of agriculture | Corine Land Cover 2006 |
| | Natural and semi-natural systems | Corine Land Cover 2006 |
| | Areas of cultural heritage | Corine Land Cover 2006, MiBACT-Italian Ministry for cultural heritage |
| | Protected areas | Corine Land Cover 2006, MATTM-Italian Ministry for Environment, Veneto Region |
| | Point and widespread sources of pollution (Directives 82/501/EC, 2008/1/EC) | ISTAT, https://prtr.eea.europa.eu |
| Flood Vulnerability (Susceptibility) | Vegetation cover | Corine Land Cover 2006 |
| | Soil type | Corine Land Cover 2006 |
| | Water height from simple gauges equipped with QR codes, which are read by technicians and citizens as well as photographs and other flood-relevant information collected via an app | Collected by AAWA |

### 3.1.2 Flood Hazard Mapping

According to Article 6 of the 2007/60/CE Flood Directive (EU, 2007), when local authorities implement a Flood Risk Management Plan, three hazard scenarios must be considered:

1. A flood with a low probability, which is 300-year return period in the study area;
2. A flood with a medium probability, which is a 100-year return period in the study area; and
3. A flood with a high probability, which is a 30-year return period in the study area.

These have been calculated using a two-dimensional hydrological and hydraulic model to generate the water levels and the flow velocities at a spatial resolution of 10 m (Ferri et al., 2010). Details of the model can be found in the Supplementary Materials. The hazard associated with these scenarios was calculated in relative terms as a value between 0 and 1.

At present, the impact of the CO is not evaluated in the hazard component as the inputs from citizens are used in real-time rather than the baseline modelling that was done to establish the areas flooded, the height and the flow velocity under three different flood return periods. In the Brenta-Bacchiglione catchment, crowdsourced observations of water level are

assimilated into the hydrological model by means of rating curves assessed for the specific river location, and directly into the hydraulic model. In the past, Mazzoleni et al. (2017) assessed the improvement of the flood forecasting accuracy obtained by integrating physical and social sensors distributed within the Brenta-Bacchiglione basin, and Mazzoleni et al. (2018) demonstrated that the assimilation of crowdsourced observations located at upstream points of the Bacchiglione catchment ensure high model performance for high lead times, whereas observations at the outlet of the catchments provide good results for short lead times.

### 3.1.3 Flood Exposure Mapping

The 2006 Corine Land Cover map provides the underlying spatial information to calculate exposure; the land use classes used here are shown in Table S2 in the Supplementary Materials. As mentioned above, the first macro-category is the people affected by the flooding, or the exposure of the population ($E_P$), which is calculated as follows:

$$E_p = F_d * F_t \tag{2}$$

where $F_d$ is a factor characterizing the density of the population in relation to the number of people present (Table 2), which uses gridded population from the census (Table 1), and $F_t$, which is the proportion of time spent in different locations (e.g., houses, schools, etc., using the land use classes listed in Table S2) over a 24 hour period (Provincia Autonoma di Trento, 2006). The four classes in Table 2 reflect a very slight decrease in exposure as population density decreases, and were defined by stakeholders in the AAWA based on guidance from ISPRA (2012). The relative values by land use class for $E_P$ are provided in Table 3.

**Table 2: A factor characterizing the density of people ($F_d$) in relation to the number of people present.**

| Number of people | $F_d$ |
|:---:|:---:|
| 1 – 50 | 0.90 |
| 51 – 100 | 0.95 |
| 101 – 500 | 0.98 |
| > 500 | 1 |

The physical exposure or impact on economic activities ($E_E$), which is the second macro-category, is calculated from the restoration costs, and the costs resulting from losses in production and services. These various costs were obtained from the Provincia Autonoma di Trento (2006) and have been calculated for each of the land use classes in Table S2. Using these costs, the relative values of $E_E$ were determined, which are listed in Table 3. Table S3 in the Supplementary Material provides a further explanation of the relation between the costs and how the relative values were derived. The final macro-category, i.e., the exposure of assets in the environmental and cultural heritage category ($E_{ECH}$), is calculated from estimates of potential damage caused by an adverse flood event. Similar to $E_E$, the costs were obtained from the Provincia Autonoma di Trento (2006) and calculated for each land use class in Table S2. The relative values of $E_{ECH}$ were then determined (listed in Table 3), where the logic behind these values is provided in Table S4 in the Supplementary Material. Note that all the relative values in Table 3 have been derived by the Provincia Autonoma di Trento (2006) from decades of experience with understanding exposure related to flood risk. Moreover, they have been tested over time and shown to be valid within AAWA. When a range of values is listed in Table 3, this reflects different types within the same land use class. For example, the value for $E_P$ ranges between 0.5-1 for industrial. This range reflects the distinction between a production cycle of 24 hours and greater, or one that is less than 24 hours. In another example, the values for $E_E$ range between 0.3-1 for specialized agriculture to reflect the distinction between crops of very low value (e.g., maize) and others of high value (e.g., vineyards). Where it was not possible to disaggregate sub-types within a given land use class, the maximum value in the range was adopted as a cautious approach.

**Table 3: The relative values of exposure for people, economic activities, and environmental/cultural assets by land use class.**

| ID | Description | $E_P$ | $E_E$ | $E_{ECH}$ |
|---|---|---|---|---|
| 1 | Residential | 1 | 1 | 1 |
| 2 | Hospital facilities, health care, social assistance | 1 | 1 | 1 |
| 3 | Buildings for public services | 1 | 1 | 1 |
| 4 | Commercial and artisan | 0.5 - 1 | 1 | 0.8 |
| 5 | Industrial | 0.5 - 1 | 1 | 0.3 - 1 |
| 6 | Specialized agricultural | 0.1 - 0.5 | 0.3 - 1 | 0.7 |
| 7 | Woods, meadows, pastures, cemeteries, urban parks | 0.1 - 0.5 | 0.3 | 0.7 |
| 8 | Tourist recreation | 0.4 - 0.5 | 0.5 | 0.1 |
| 9 | Unproductive | 0.1 | 0.1 | 0.3 |
| 10 | Ski areas, Golf course, Horse riding | 0.3 - 0.5 | 0.3 - 1 | 0.3 |
| 11 | Campsites | 1 | 0.5 | 0.1 |
| 12 | Roads of primary importance | 0.5 | 1 | 0.2 |
| 13 | Roads of secondary importance | 0.5 | 0.5 - 1 | 0.1 |
| 14 | Railway area | 0.7 - 1 | 1 | 0.7 |
| 15 | Area for tourist facilities, Zone for collective equipment (supra-municipal, subsoil) | 1 | 0.3 | 0.3 |
| 16 | Technological and service networks | 0.3 - 0.5 | 1 | 0.1 |
| 17 | Facilities supporting communication and transportation networks (airports, ports, service areas, parking lots) | 0.7 - 1 | 1 | 1 |
| 18 | Area for energy production | 0.4 | 1 | 1 |
| 19 | Landfill, Waste treatment plants, Mining areas, Purifiers | 0.3 | 0.5 | 1 |
| 20 | Areas on which plants are installed as per Annex I of Legislative Decree 18 February 2005, n. 59 | 0.9 | 1 | 1 |
| 21 | Areas of historical, cultural and archaeological importance | 0.5 - 1 | 1 | 1 |
| 22 | Environmental goods | 0.5 - 1 | 1 | 1 |
| 23 | Military zone | 0.1 - 1 | 0.1 - 1 | 0.1 - 1 |

### 3.1.4 Flood Vulnerability Mapping

Vulnerability is also quantified for each of the three macro-categories (i.e., people, economic activities and environmental/cultural-archaeological assets affected) as outlined below but we additionally differentiate between physical and social vulnerability as described in Section 3.1.

**(i) Physical vulnerability of people affected by flooding**

The physical vulnerability associated with people considers the values of flow velocity ($v$) and water height ($h$) that produce "instability" with respect to remaining in an upright position. Many authors have dealt with the instability of people in flowing water (see e.g., Chanson and Brown, 2018), and critical values have been derived from the product of $h$ and $v$. For example, Ramsbottom et al. (2004) and Penning-Rowsell et al. (2005) have proposed a semi-quantitative equation that links a flood hazard index, referred to as the Flood Hazard Rating (FHR), to $h$, $v$ and a factor related to the amount of transported debris, i.e., the Debris Factor (*DF*), as follows:

$$FHR = h * (v + 0.5) + DF \tag{3}$$

The values of the *DF* related to different ranges of *h*, *v* and land use are reported in Table 4, which were taken from a study by the UK Department for Environment, Food and Rural Affairs (DEFRA) and the UK Environment Agency (2006) as reported in ISPRA (2012).

**Table 4: The Debris Factor (*DF*) for different water heights (*h*), flow velocities (*v*) and land uses. Source: ISPRA (2012), with**
280 **reference to DEFRA and UK Environment Agency (2006)**

| Values of *h* and *v* | Grazing/Agricultural land | Forest | Urban |
|---|---|---|---|
| 0 m < *h* ≤ 0.25 m | 0 | 0 | 0 |
| 0.25 m < *h* ≤ 0.75 m | 0 | 0.5 | 1 |
| *h* > 0.75 OR *v* > 2 m/s | 0.5 | 1 | 1 |

Using the FHR, the physical vulnerability of the population can be calculated, which is summarized in Figure 4. These three values of Vp were proposed in the ISPRA (2012) guidelines.

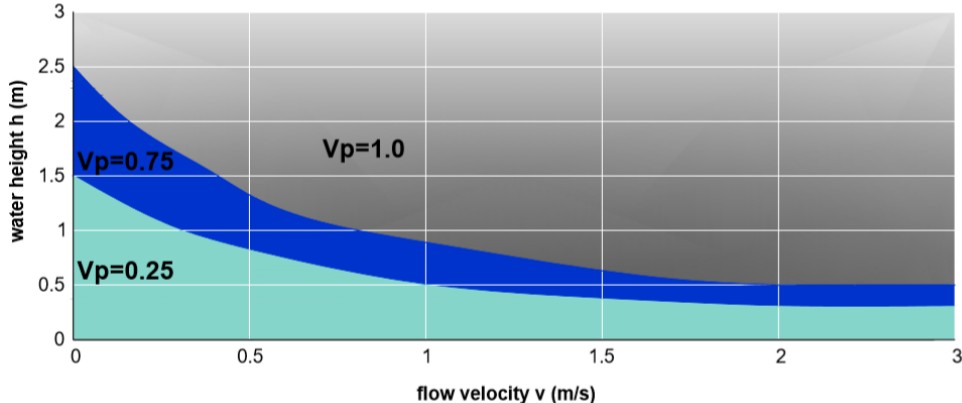

**Figure 4: Physical vulnerability (Vp) values for the population as a function of water height (*h*) and flow velocity (*v*).**

**(ii) Social vulnerability of people affected by flooding**

Figure 5 shows the components of social vulnerability, i.e., the adaptive and coping capacity and their respective indicators, along with the weights associated with each of them.

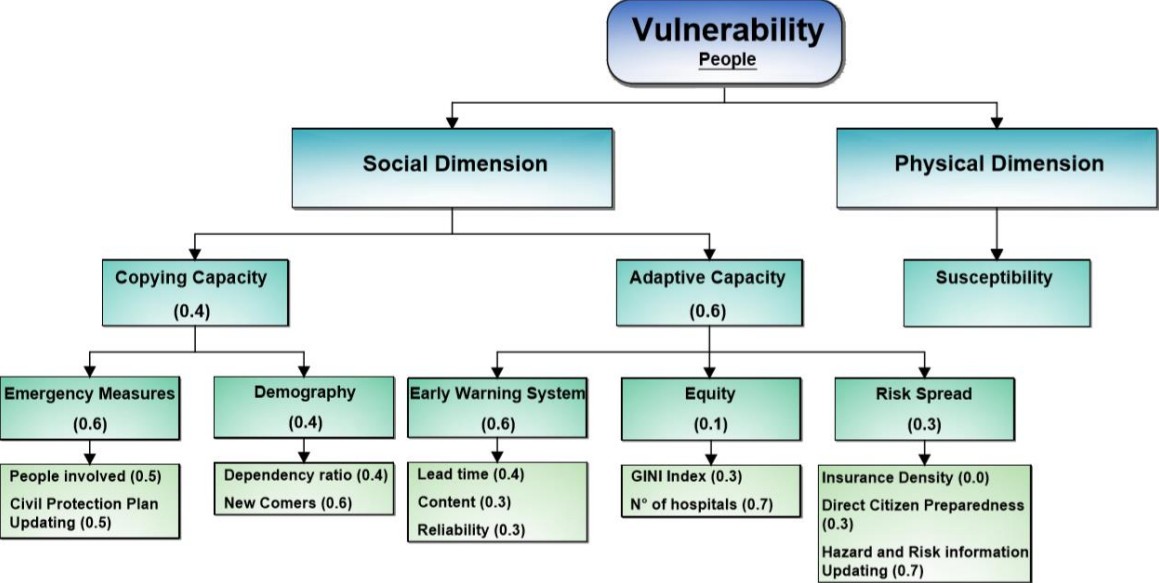

**Figure 5: Hierarchical combination of indicators and relative weights (in brackets) to calculate the vulnerability of the population.**

The weights and values assigned to each of these indicators have been determined through an expert consultation process carried out by AAWA. Because the different indicators have varying units of measurement, they were first normalized so that they could be combined. Several normalization techniques exist in the literature (Biausque, 2012) but the 'value function' was chosen because it represents a mathematical expression of a human judgement that can be compared in a systematic and explicit way (Beinat, 1997; Mojtahed, et al., 2013). The principal aim of this consultation process was to assign a value between 0 and 1 to people's vulnerability, considering the relative weight of each indicator. The stakeholders engaged were the members of the Technical Committee of the water basin authority made up of technical representatives of the regional and provincial administrations belonging to the Eastern Alps District, as well as experts from the professional and academic sectors (i.e., around 20 people). The process to identify the weights started with several discussions, the results of which were interpreted and translated into values/weights by AAWA, who then re-proposed these values to the experts, obtaining their consensus. Similarly to what was done to identify the weights, AAWA formulated an internal study for the definition of the value functions for the different indicators, which were then proposed and discussed with the members of the Technical Committee, obtaining their consensus.

The coping capacity is comprised of the following demographic and emergency measure indicators, where the corresponding value functions are shown in Figure S1:

- Dependency ratio: the number of citizens aged under 14 and over 65 as a percentage of the total population. A high value of this index implies a reduced ability to adapt to hazardous events.

- Foreigners: the number of foreigners as a percentage of the total population. Due to language barriers and other cultural reasons, areas with a high number of immigrants may not cope as well after a flood event and during emergency situations.

- Number of people involved in emergency management: the number of operators who have been trained to manage an emergency in the region, expressed qualitatively as low, medium and high.

- How frequently civil protection plans are updated: Updating is measured in months to years and indicates how often new hydraulic, urban and technological information is incorporated into civil protection plans.

The adaptive capacity is comprised of three components: the early warning system, equity and risk spread. Early warning systems are evaluated according to three criteria, where the value functions are shown in Figure S2:

- Lead time (or warning time): the number of hours before an event occurs that was predicted by the early warning system.

- Content: the amount of information provided by the early warning system, such as the time and the peak of the flooding at several points across the catchment.

- Reliability: this is linked to the uncertainty of the results from the meteorological forecasts and the hydrological models (Schroter et al., 2008). False alarms can cause inconvenience to people, hinder economic activities, and people may be less likely to take warnings seriously in the future; therefore, they should be minimized.

Finally, equity and spread (shown in Figure S3) are characterized by:

- Gini Index: a measure of the inequality of income distribution within the population. A value of 0 means perfect equality while 1 is complete inequality.

- Number of hospital beds: this is calculated per 1000 people.

- Insurance density: this is the ratio of total insurance premiums (in €) to the total population (Lenzi and Millo, 2005). Values with higher insurance density lead to increased adaptive capacity. However, the insurance density is set to zero because insurance companies in this part of Italy do not currently offer premiums to protect goods against flood damage.

- The frequency at which information on hazard and risk are updated: this is measured in months to years and indicates the ability of institutions to communicate the conditions of danger and risk to the population.

- Involvement of citizens: This is based on the number of students, associations such as farmers and professionals, and citizens that can be reached across large areas through social networks (WP7 WSI Team, 2013) to disseminate information. Figure S3d shows the maximum achievable value in the different categories of citizen involvement.

The value for social vulnerability is the sum of the coping and adaptive capacities while the final value for the vulnerability of people is calculated by multiplying the physical and the social vulnerability together.

**(iii) Physical vulnerability of economic activities affected by flooding**

The vulnerability associated with economic activities considers buildings, network infrastructure and agricultural areas. For buildings, the effects from flooding include collapse due to water pressure and/or undermining of the foundations. Moreover, solid materials, such as debris and wood, can be carried by a flood and can cause additional damage to structures. A damage function for brick and masonry buildings has been formulated by Clausen and Clark (1990). Laboratory results have shown that at a water height of 0.5m, the loss to indoor goods is around 50%, which is based on an evaluation made by Risk Frontiers, an independent research center sponsored by the insurance industry. The structural vulnerability of buildings and losses of associated indoor goods is shown in Figure S4 as a function of the height of the water and flow velocity, which are applied to land use types containing buildings (Table S2). For the camping land use type 11 (Table S2), the values have been modified based on results from Majala (2001).

Vulnerability of the road network is evaluated for land use types 12 and 13 in Table S2, which occurs when it is not possible to use the road due to flooding. This is based on an estimation of the water height and the critical velocity at which vehicles become unstable during a flood, which are derived from direct observation in laboratory experiments and from a report on the literature in this area (Reiter, 2000; Shand et al., 2011); the vulnerability function for the road network is presented in Figure S5. Regarding technological and service networks (land use type 16, Table S2), we assume a vulnerability value equal to 1 if the water height and flow velocity are greater than 2 m and 2 m/s, respectively, otherwise 0.

To assess the vulnerability in agricultural areas (land use types 6 and 7 in Table S2), we assume that the damage is related to harvest loss, and when considering higher flow velocities and water heights, to agricultural buildings and internal goods. Citeau (2003) provides relationships that take water height and flow velocity into account, e.g., the maximum height is 1 m for orchards and 0.5 m for vineyards, and the maximum velocity varies from 0.25 m/s for vegetables and 0.5 m/s for orchards. Concerning cultivation in greenhouses, the maximum damage occurs at a height of 1 m. Finally, high velocities can cause direct damage to cultivated areas but can also lead to soil degradation due to erosion. The vulnerability values for

four different types of land as a function of water height and flow velocity are shown in Figure S6. In the case of unproductive land (land use type 9 in Table 1), the vulnerability is assumed to be 0.25, regardless of the $h$ and $v$ values.

**(iv) Physical vulnerability of environmental and cultural heritage assets affected by flooding**

Environmental flood susceptibility is described using contamination/pollution and erosion as indicators. Contamination is caused by industry, animal/human waste and stagnant flooded waters. Erosion can produce disturbance to the land surface and to vegetation but can also damage infrastructure. The approach taken here was to identify protected areas that could potentially be damaged by a flood. For areas susceptible to nutrients, including those identified as vulnerable in Directive 91/676/CEE (Nitrate), and for those defined as susceptible in Directive 91/271/CEE (Urban Waste), we assume a value of 1 for vulnerability (land use type 20 in Table S2). Similarly, in areas identified for habitat and species protection, i.e., sites belonging to the Natura 2000 network established in accordance with the Habitat Directive 92/43/CEE and Birds Directive 79/409/CEE (land use types 8 and 22 in Table S2), the presence of relevant pollution sources was identified (Tables 1 and S1) and assigned a vulnerability of 1. In the absence of pollution sources, the vulnerability was calculated as 0.25 if the flood velocity was less than or equal to 0.5 m/s and the water height was less than or equal to 1 m; otherwise it was 0.5. Regarding cultural heritage (land use type 21 in Table S2), we assigned a vulnerability of 1 to these areas.

**3.1.5 Mapping flood risk before and after implementation of a CO on flood risk management**

Once the hazard, exposure and vulnerability are mapped, the flood risk, $R$, for the three flood hazard scenarios, $i$, can be mapped as follows:

$$R = \sum_{i=1}^{3} R_i = \frac{w_P\,(H_i \cdot E_P \cdot V_P) + w_E\,(H_i \cdot E_E \cdot V_E) + w_{ECH}\,(H_i \cdot E_{ECH} \cdot V_{ECH})}{w_P + w_E + w_{ECH}} \tag{4}$$

where $H$, $E$ and $V$ are the hazard, exposure and vulnerability associated with the three macro-categories $P$, $E$ and $ECH$ are the people, economic activities and environmental/cultural-archaeological assets affected, and $w_P$, $w_E$ and $w_{ECH}$ are weights applied to each macro-category, with values of 10, 1 and 1, respectively, which were defined based on stakeholder interviews undertaken by AAWA. To establish the level of risk, four risk classes were defined (Table 5).

Table 5: Definition of risk classes.

| Range of R | Description | Risk Category |
|---|---|---|
| 0.1 < R ≤ 0.2 | Low risk where social, economic and environmental damage are negligible or zero | R1 |
| 0.2 < R ≤ 0.5 | Medium risk for which minor damage to buildings, infrastructure and environmental/cultural heritage is possible, which does not affect the safety of people, the usability of buildings or economic activities | R2 |
| 0.5 < R ≤ 9 | High risk in terms of safety of people, damage to buildings and infrastructure (and/or unavailability of infrastructure), interruption of socio-economic activities and damage related to environmental/cultural heritage | R3 |
| 0.9 < R ≤ 1 | Very high risk including loss of human life and serious injuries to people, serious damage to buildings, infrastructure and environmental/cultural heritage, and total disruption of socio-economic activities | R4 |

These risk classes were then mapped with and without the implementation of the CO for flood risk management. The main change in the calculation of risk is in the social dimension of vulnerability. Before the CO was implemented, this component had a value of 0.9. Based on the experience gained in the WeSenseIt project and the goals of the CO, the changes in social vulnerability with the implementation of the CO are shown in Table 6, which decreases the social vulnerability to a value of 0.63. For example, in the coping capacity, the number of people employed in emergency management does not change but as

a result of the CO, they will work in a much more efficient manner due to the technology that allows for better emergency management. In terms of content of the Early Warning System, with the CO, very detailed information will be obtained, further enriched by citizen reports (including reports from waterways that were not previously equipped with measuring instruments) and by a monitoring network that will be equipped with a further eight thermo-pluviometric stations, 12 hydrometric stations (equipped with a double transmission system), and 58 hydrometric and six snow measuring rods. The forecasted water level is available at every section of the Brenta-Bacchiglione River system. Hence, the content will be enhanced through the implementation of the CO. The reliability of the Early Warning System increases to very high due to the involvement of trained citizens who provide information and sensor readings that are used to validate and feed the hydrological/hydraulic model (i.e., the data assimilation module). The assumption was made based on the results obtained in Mazzoleni et al. (2017, 2018) and by considering a hypothetical situation in which a widely distributed crowdsourcing data acquisition process is in place due to the expected high level of citizen engagement. These tools will also lead to more frequent updating of civil protection plans as well as hazard and risk information updates. In addition, the early warning system will improve in terms of lead time, content and reliability through the greater involvement of trained volunteers and citizens.

**Table 6: Changes in the indicators of social vulnerability with and without implementation of the CO on flood risk management.**

| Social vulnerability | Indicator | Value without CO | Value with CO |
|---|---|---|---|
| Adaptive capacity | Number of people involved in emergency management | Medium | High |
| | Frequency of civil protection plan updating | > 5 years | > 2 years |
| Coping capacity | Lead time of Early Warning System | < 6 hours | 24-72 hours |
| | Content of Early Warning System | Little information | Very detailed information |
| | Reliability of Early Warning System | None | High |
| | Citizen involvement | None | Citizens of large area |
| | Hazard and risk information updating | > 5 years | 1-2 years |

**3.2 Financial quantification of the direct damage due to flooding with and without implementation of a flood risk management CO**

To estimate the direct tangible costs due to damage resulting from a flood event, we use the maximum damage functions related to the 44 land use classes in the Corine Land Cover developed by Huizinga (2007) for the 27 EU member states, which are based on replacement and productivity costs and their gross national products. The replacement costs for damage to buildings, soil and infrastructure assume complete rebuilding or restoration. Productivity costs are calculated based on the costs associated with an interruption in production activities inside the flooded area. The maximum flood damage values for the EU-27 and various EU countries are provided in Table S5. The direct economic impact of the flood is calculated by multiplying the maximum damage values per square meter (in each land use category) by the corresponding areas affected by the floods, i.e., the flood hazard (Section 3.1.2), weighted by the vulnerability value associated with each grid cell. Since the land use map used in this study does not distinguish between industrial and commercial areas, the average of the respective costs per square meter (475.5 €/m$^2$) has been applied. Moreover, in discontinuous urban areas, 50% of the value of the damage related to continuous urban areas (i.e., 309 €/m$^2$) was applied, due to the lower density of buildings in these areas.

The average annual expected damage (*EAD*) can be calculated as follows, where *D* is the damage as a function of the probability of exceeding *P* for a return time *i* (Meyer et al., 2007):

$$EAD = \sum_{i=1}^{k} \frac{D(P_{i-1})+D(P_i)}{2} \cdot |P_i - P_{i-1}| \tag{5}$$

$$D(P_i) = \sum_i \frac{\sum_j A_{Dj}^i {}^{*} w_{Dj}}{\sum_j w_{Dj}} \cdot D^i \tag{6}$$

where $w_{Dj}$ is the weight of the damage class, $j$ is the damage category and $D$ is the damage value shown in Table S5. The EAD is calculated before and after implementing the CO for flood risk management. The monetary benefits are the "avoided" damage costs (to people, buildings, economic activities, protected areas, etc.) if the CO for flood risk management is implemented.

## 4 Results

**4.1 Hazard and flood risk estimates before and after implementation of a flood risk management CO**

The results of the numerical simulations from the hydraulic model, which were carried out based on the methodology described in the Supplementary Materials, have shown that in some sections of the Bacchiglione River, the flow capacity will exceed that of the river channel. This will result in flooding, which will affect the towns of Torri di Quartesolo, Longare and Montegaldella. There will also be widespread flooding in the cities of Vicenza and Padua, including some industrial

areas and others rich in cultural heritage. For a 30-year flood event, the potential flooding could extend to around 40,000 ha, where 25% of the area contains important urban areas with significant architectural assets. In the case of a 100-year flood event, the areas affected by the flood waters increase further, with more than 50,000 ha flooded, additionally affecting agricultural areas. The results of the simulations are summarized in Tables 7 and 8 in terms of the areas affected in the catchment for different degrees of hazard and risk for 30-, 100- and 300-year flood events.

**Table 7: The hazard classes for each return period in terms of area flooded.**

| Hazard class | 30 year return period | 100 year return period | 300 year return period |
|---|---|---|---|
| | Area (km$^2$) | | |
| Low | 185.12 | 294.77 | 370.07 |
| Medium | 118.87 | 161.82 | 225.67 |
| High | 54.18 | 74.55 | 104.61 |
| Total | 358.17 | 531.14 | 700.35 |

**Table 8: The risk classes for each return period in terms of area flooded (km$^2$) before and after implementation of the CO.**

| Risk Class | Before implementation of the CO | | | After implementation of the CO | | |
|---|---|---|---|---|---|---|
| | 30 year return period | 100 year return period | 300 year return period | 30 year return period | 100 year return period | 300 year return period |
| Low (R1) | 160.29 | 254.29 | 318.80 | 170.96 | 268.68 | 337.78 |
| Medium (R2) | 137.26 | 191.89 | 262.03 | 168.99 | 235.18 | 322.41 |
| High (R3) | 56.70 | 79.23 | 110.29 | 18.19 | 27.19 | 40.04 |
| Very High (R4) | 3.92 | 5.73 | 9.23 | 0.03 | 0.09 | 0.12 |
| Total | 358.17 | 531.14 | 700.35 | 358.17 | 531.14 | 700.35 |

Figure 6 shows the areas at risk in the territory of Padua for a 100-year flood event before implementation of a CO on flood risk management. Risk classes R1 (low risk) and R2 (medium risk) have the highest areas for all flood event frequencies. Although areas in R3 (high risk) and R4 (very high risk) may comprise a relatively smaller area when compared to the total area at risk, these also coincide with areas of high concentrations of inhabitants in Vicenza and Padua.

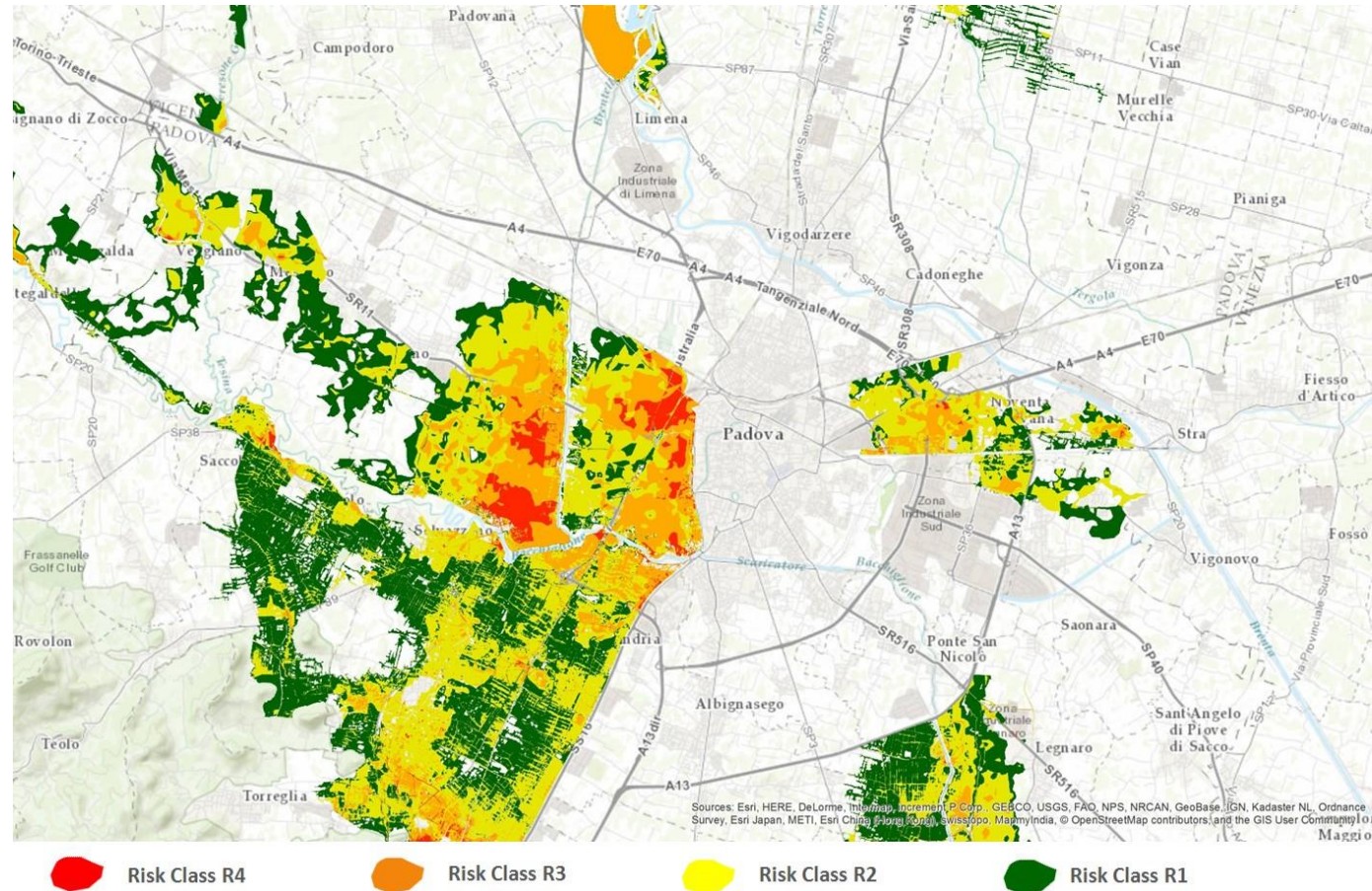

**Figure 6: Risk map for the metropolitan area of Padua for a 100-year flood event before implementation of a CO on flood risk management. Sources: Esri, HERE, DeLorme, Intermap, Increment P. Corp., GEBCO, USGS, FAO, NPS, NRCAN, GeoBase, IGN, Kadaster NL, Ordnance Survey, Esri Japan, METI, Esri China HongKong, swisstopo, MapmyIndia, © OpenStreetMap contributors 2020. Distributed under a Creative Commons BY-SA License, and the GIS User Community**

After implementation of a CO for flood risk management, the flood risk is reduced (Table 8) due to the reductions in vulnerability outlined in section 3.1.5. The areas affected in the high (R3) and very high classes (R4) are significantly reduced (R4 to almost zero) but the areas in the lower risk classes increase. This occurs because the total area affected by the flood hazard is the same before and after implementation of a CO. What changes is the distribution between risk classes, i.e.,
 R3 and R4 are reduced, which means that the areas at risk in classes R1 and R2 will increase.The risk map for a 100-year flood event for the territory of Padua is shown in Figure 7, where the reduction in areas at high and very high risk are clearly visible compared to the situation before implementation of the CO (Figure 6).

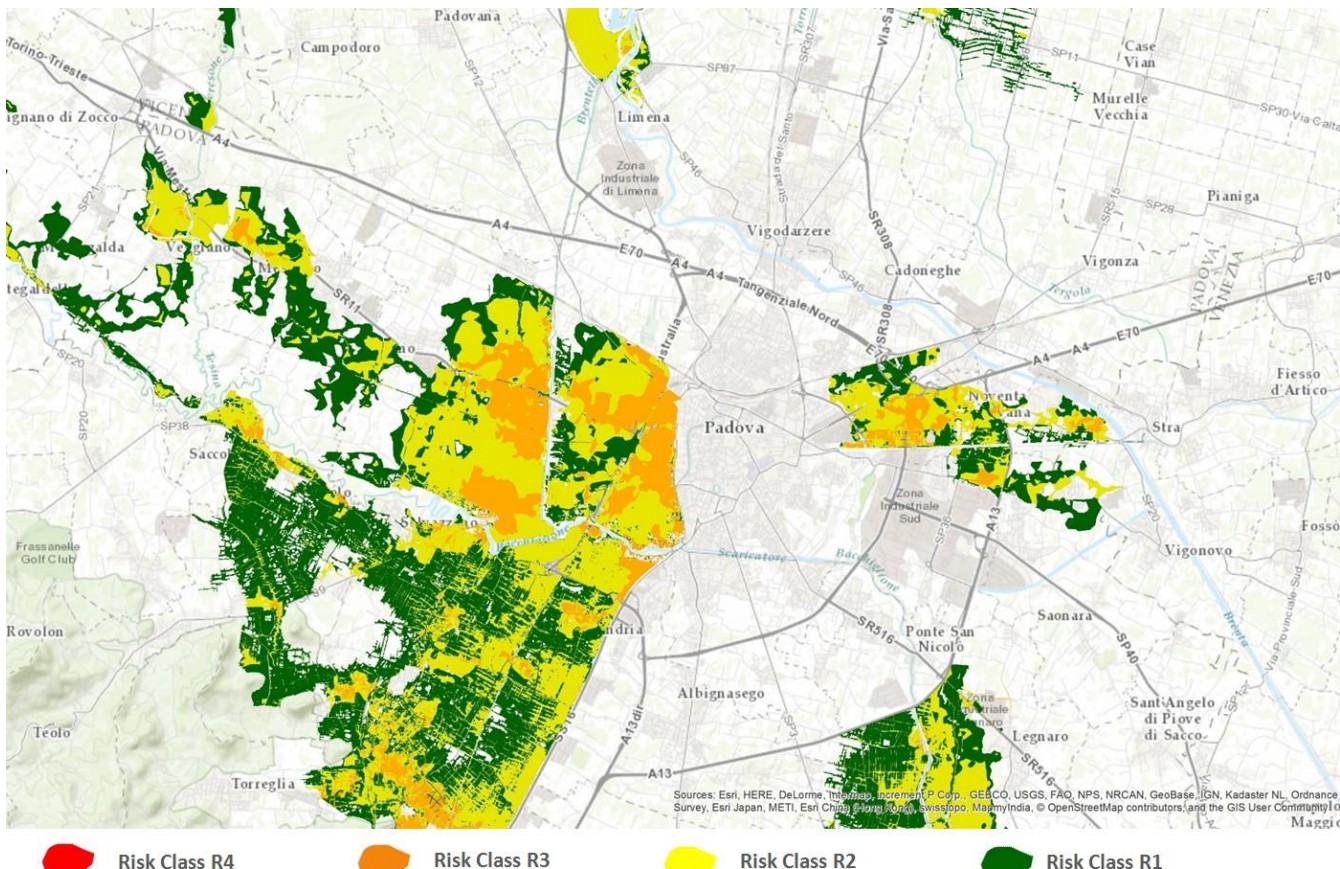

| | Risk Class R4 | | Risk Class R3 | | Risk Class R2 | | Risk Class R1 |

**Figure 7: Risk map for the metropolitan area of Padua for a 100-year flood event after implementation of a CO on flood risk management. Sources: Esri, HERE, DeLorme, Intermap, Increment P. Corp., GEBCO, USGS, FAO, NPS, NRCAN, GeoBase, IGN, Kadaster NL, Ordnance Survey, Esri Japan, METI, Esri China HongKong, swisstopo, MapmyIndia, © OpenStreetMap contributors 2020. Distributed under a Creative Commons BY-SA License, and the GIS User Community**

### 4.2 Expected damage with and without implementation of a flood risk management CO

The direct damage was calculated for the three flood scenarios: high chance of occurrence (every 30 years), medium (every 100 years) or low (every 300 years), which is summarized in Table 9. In the event of very frequent flood events, urban areas will be damaged. Furthermore, moving from an event with a high probability of occurrence to one with a medium probability results in a significant increase in the area flooded (i.e., a 48% increase as shown in Table 8) but with a smaller increase in damage (i.e., around 20%). This is explained by the fact that the flooded areas in a 100-year flood event (but not

present in a 30-year flood event) are under agricultural use. Similar patterns can be observed when comparing floods with a low and high probability of occurrence. Substituting the values in Table 9 into equation (5), we obtain an expected average annual damage (EAD) of 248.5 million Euros. The residual damage was then calculated for the three flood scenarios after implementation of the CO on flood risk reduction, which is shown in Table 9. Substituting these residual damage values into equation (5), we obtain an EAD of 111.3 million Euros, which is a 45% reduction in the damage compared to results without

implementation of the CO.

The CO for flood risk management has an estimated cost of around 5 million Euros (as detailed in Table S6 in the Supplementary Materials), after which it will be evaluated and further funding sought. Taking the EAD with and without implementation of the CO, the annual benefit in terms of avoided damage is approximately 137.2 million Euros. Hence the benefits considerably outweigh the costs. The same methodology was applied to the construction of a retention basin in the

490 municipalities of Sandrigo and Breganze (in an independent exercise) to improve the hydraulic safety of the Bacchiglione River. Against an expected cost of 70.7 million Euros, which is much higher than the estimated cost for implementing the CO, a significant reduction in flooded areas would be obtained although high risk would still be evident in the city of Padua.

In terms of damage reduction with the construction of the retention basin, we would obtain an EAD of 140.7 million Euros so the cost to benefit ratio would be much lower.

Table 9: Comparison of the direct (without CO) and residual damage (with CO) for three flood scenarios and the cost difference.

| Scenarios (chance of flood occurrence) | Return period | Direct damage (million €) | Residual damage (million €) | Difference in costs (million €) |
|---|---|---|---|---|
| High | 30 years | 7,053 | 1,573 | -5,480 |
| Medium | 100 years | 8,670 | 5,440 | -3,230 |
| Low | 300 years | 10,853 | 3,420 | -7,433 |

## 5 Discussion and Conclusions

There is currently a lack of available, appropriate and peer-reviewed evaluation methods and evidence on the added value of citizen observatories, which is required before they will be more widely adopted by policy makers and practitioners. This

paper has aimed to fill this gap by demonstrating how a traditional cost-benefit analysis can be used to capture the value of a CO for flood risk management. Although the CO is still being implemented, the proposed methodology was applied using primary empirical evidence from a CO pilot that was undertaken by the WeSenseIt project in the smaller Bacchiglione catchment to guide changes in the values associated with social vulnerability once the CO is implemented. This allowed the risk and flood damages to be calculated with and without implementation of the CO, which showed that implementation of a

CO in the Brenta-Bacchiglione catchment is able to reduce the damage, and consequently the risk, for the inhabited areas from an expected average annual damage (EAD) of €248.5 to €111.3 million euros, i.e., a reduction of 45%. Hence, the implementation of the CO could significantly reduce the damage and consequently the risk for the inhabited areas of Vicenza, Padua, Torri di Quartesolo, Longare and Montegaldella. The nature of the methodology also means that it can be applied to other catchments in any part of Italy or other parts of the world that are considering the implementation of a CO

for flood risk management purposes.

The main impact of the CO on flood risk management has been to lower the social vulnerability of risk, both in terms of adaptive as well as coping capacity. This finding is consistent with other studies of citizen science that aim to capture the impacts on social vulnerability. Bremer et al. (2019) in their case study in Bangladesh found that citizen science has had a high impact on adaptive capacity in terms of individual awareness and understanding of local rainfall, learning that they

applied in adaptive practices at work and at home, as well as local leadership. Improvements in social capital (trust, sharing experience and formal/informal interactions) were also measurable. This provides support for the argument that CO impacts, especially capacity-related ones, do not necessarily (have to) materialise (only) via formal policy mechanisms. Both coping and adaptive capacity have individual, community as well as policy dimensions, not all of which are impacted in parallel nor to the same degree; moreover, adaptive capacities are context specific.

Regarding the impact on estimating other flood risk drivers, at present, the impact of citizens is not evaluated in the hazard component as the inputs from citizens would be used in real-time rather than the baseline modelling that was done to establish the areas flooded, the height and the flow velocity under three different flood return periods. Instead, the contribution of citizens is incorporated into the Early Warning System component of social vulnerability through improvements in the reliability, lead time and information content of the system (Figure S2 in the Supplementary

Information) as well as components of Adaptive Capacity (Hazard and risk information updating and Citizen involvement - Figure S3 in the Supplementary Information). Similarly, there is currently no impact of citizens/experts on exposure or physical vulnerability as this analysis is based on land use categories rather than individual buildings, where in the latter it might be possible to capture small changes done at the household, building or feature level. However, this is not part of the current methodology. A second aspect is increased awareness/participation in combination with data provisioning, i.e., the

app provides information about flood risk to citizens while at the same time asking for inputs/participation that can be used

to feed the model and/or, in real-time, to provide information to help emergency response. The concept of a CO is built on the idea of two-way communication between the citizens/experts and the local authorities.

We do acknowledge that this methodology is built on many assumptions, i.e., the numerous coefficients, value functions and weights used to estimate the exposure and vulnerability. We have summarized these assumptions in Table S1 of the Supplementary Material. Many of these values have been derived through expert consultation and experience, and they been validated internally within AAWA or by other Italian agencies. Value functions, in particular, are a way of capturing human judgement in way that can be quantified in situations of high uncertainty. We would argue that the expert consultations have not been undertaken lightly and have often resulted in conservative estimates in the values. We have tried to reflect this in Table S1. Other values have been derived from the literature, all of which will have some uncertainties associated with their derivation. The primary objective of the paper was never to do a fully-fledged uncertainty analysis but to present a methodology that could be shared with experts, and local and national authorities, to evaluate the potential of a CO solution in monetary terms with regards to reducing the vulnerability of flood risk. The weights adopted and the assumptions made, which depend on the policies and the local context of the study area, do not affect the value of the method presented, which can be applied to other river basins with the adoption of different weights. That said, this cost-benefit analysis is hypothetical because the CO for flood risk management is still being implemented. Hence the real benefits will only be realized once the CO is fully operational. Our goal will then be to verify the assumptions and the empirical weight factors adopted, via a more detailed quantitative analysis.

Another limitation of the analysis presented here is that we did not consider indirect costs, such as those incurred after the event takes place, or in places other than those where the flooding occurred (Merz et al., 2010). In accordance with other authors (e.g., van der Veen et al., 2003), all expenses related to disaster response (e.g., costs for sandbagging, evacuation) are classified as indirect damage. However, the presence of the CO in this catchment does reduce the costs related to emergency services, securing infrastructure, sandbagging and evacuation, all of which can be substantial during a flood event. Although the people involved in the emergency services are the same, they are employed in a much more efficient way as a result of the technology developed with the CO, which allows for better management of the teams responding to the event and the efficient assignment of tasks based on an operator's location. Therefore, an analysis that takes indirect costs into account could help to further convince policy makers of the feasibility of a CO solution. Similarly, intangible costs were not considered, i.e., the values lost due to an adverse natural event where monetary valuation is difficult because the impacts do not have a corresponding market value (e.g., health effects). Furthermore, the vulnerability assessment of economic activities considers only water depth and flow velocity but not additional factors such as the dynamics of contamination propagation in surface waters during the flood or the duration of the flood event, all of which could be taken into account in estimating the structural damage and monetary losses in the residential, commercial and agricultural sectors.

Despite these various limitations, this analysis has highlighted the feasibility of a non-structural flood mitigation choice such as a CO for flood risk management compared to the implementation of much more expensive structural measures (e.g., retention areas) in terms of the construction costs and the cost of maintenance over time. The evidence on the costs and benefits of COs for flood risk management generated by this case study can provide insights that policy makers, authorities and emergency managers can use to make informed choices about the adoption of COs for improving their respective flood risk management practices. In Italy, in general, citizen participation in flood risk management has been relatively limited. By involving citizens in a two-way communication with local authorities through a CO, flood forecasting models can be improved, increased awareness of flood hazard and flood preparedness can be achieved, and community resilience to flood risk can be bolstered. The previous strategy in the Brenta-Bacchiglione catchment has focused on structural flood mitigation measures, dealing with emergencies and optimizing resources for rapid response. The inclusion of a CO on flood risk management has been a true innovation in the flood risk management strategies of this region, which can also be transferred to other catchments. There are plans to extend the CO to other basins in the Eastern Alps, which are similar in size and

hydrological characteristics. These are complex hydrographic basins with very variable regimes, from rapid
response/torrential rainfall events of the alpine territories to the alluvial plain, which is composed of mountain and lowland
river networks, artificial networks of reclamation and natural and/or artificial reservoirs. In general, this methodology can be
applied to catchments larger than 100 km$^2$ where model forecasts would be most useful.

Future research will focus on validating the results once the CO is operational as well as application of the methodology
in other catchments and to other fields of disaster management beyond floods. Such applications will serve to generate a
broader evidence base for using these types of cost-benefit methodologies to justify the implementation of COs.

**Acknowledgements**

This research was supported by the EU's FP7 WeSenseIt project (No. 308429) and received funding from the EU's 2020
research and innovation programme under the WeObserve (No. 776740) and LandSense (No. 689812) projects.

**Author contributions**

All authors contributed to the writing and revisions of the paper. Michele Ferri and Martina Monego undertook the
hydrological-hydraulic modelling and the cost-benefit analysis.

**Competing interests**

The authors declare that there are no competing interests.

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
