# Peer review of "The Value of Citizen Science for Flood Risk Reduction: Cost-benefit Analysis of a Citizen Observatory in the Brenta-Bacchiglione Catchment"

_Hydrology and Earth System Sciences, 2020_

## Referee Comment (RC1) · Anonymous Referee #1 · 3 Aug 2020

General comments Ferri et al. assess the flood risk and related costs in the Brenta-Bacchiglione catchment in Italy to evaluate the contribution of the establishment of a citizen observatory to flood risk mitigation. The authors also use this case study to demonstrate the validity of cost-benefit analysis to assess the value of citizen observatories in flood risk mitigation. As citizen science is a fairly 'hot topic' in hydrology at the moment, I think this is a timely study, providing a relevant tool that can be applied in flood risk management. The manuscript is well-written and fits well within the scope of HESS. I do, however, have a few questions and comments that I would like the authors

to address.

In the introduction you mention that several studies identified that the link of COs to authorities and policy does not necessarily lead to increased participation or improved participation. Yet, in the cost-benefit analysis with CO, you assume a positive impact of the CO on numerous social vulnerability indicators based only on the outcomes of the pilot study. I understand that the focus of this manuscript is to demonstrate the use of a cost-benefit analysis in this context, but it would nevertheless be interesting to discuss how citizen science or CO projects in other regions affected these social vulnerability indicators. This would also put the results of this study in a broader context, which is currently missing.

This brings me to another point. From the methods, results and discussion I got the impression that most of the benefit of the citizen observatory came from the increased awareness and participation rather than just data provisioning. In the introduction, the role of citizen science and COs in data collection is actually highlighted. Also in Section 2.3, where the CO in the Brenta-Bacchiglione catchment is described, the role of 'experts' and citizens seems to focus on data acquisition, whereas the impact on estimating other flood risk drivers has not been explained as much. If you could elaborate on how the CO contributes to these aspects, this would support the (rather many) assumptions made within this study. In addition, it would be interesting to discuss at some point in the manuscript how the additional data (especially water levels) could contribute to improved hazard evaluation in your case study.

More specific comments and requests for further clarification on certain points in the manuscript are provided below.

Specific comments L. 19-20: I would use citizen observatories in this sentence as well, since your manuscript evaluates how these can contribute to risk reduction.

Figure 1: Please add legend to map and clearly indicate the boundaries of the Brenta-Bacchiglione catchment.

L. 143-145: What is the sustainability of such an arrangement, whereby the technicians get paid for each trip, once the project ends? Where would the funds come from?

L. 151-159: Could you be more specific on the kind of observations that citizens can contribute? I would imagine these are less 'complicated' than the contributions of the trained volunteers and technicians. Furthermore – as mentioned in the general comments – how will citizen engage further in risk reduction such that flood risks can be reduced?

L. 185: Supplementary tables: Please change the numbers of the supplementary tables in such a way that they are numbered according their appearance in the text.

L. 223-227: The section on EE and EECH is quite unclear and could do with some more detail, as I cannot really follow what has been done based on the current description.

Table 3: Are these relative values based on the data explained in the previous paragraphs? This seems straightforward for EP, but it is not clear how EE and EECH translate from costs to a relative value. And how have these values been tested and shown to be valid?

Figure 3: I would place this figure in Section (ii), where you explain how the weights and values have been obtained. How did you define the indicators? Is this based on existing literature? And could you provide more information on the stakeholder engagement to identify weights and values? Which stakeholders were engaged and how were the weights and values determined? Average of what different stakeholders provided? Or did certain stakeholder provide info for certain indicators only?

Table 4: Include the references to your data sources in the table (either in the caption or as footnote) if the values are directly taken from the studies you mention in L. 252-254.

Figure 4 and similar figures: Are only the values 0.25, 0.75 and 1.0 included in the analysis? If so, why particularly these values and not values in between?

L. 266-268: How were the value functions for the different indicators defined? Was this

also part of the stakeholder consultation?

L. 284-288: How were content and reliability quantified? Simply assigning it to one of the categories shown in Figure S2 could be quite subjective.

L. 299-302: More information on this is required. From what is provided, it is difficult to understand how to interpret Figure S3d. What does no category mean? That no citizens are involved (which would correspond to zero citizen involvement and thus increased vulnerability)?

L. 318-319: In the caption of Figure S3, it says these values are for network infrastructure. Does this only apply to roads? If so, please change accordingly.

L. 356-358: If social vulnerability decreases to 0.63, what was its original value?

Tables 8-11: If you combine the tables, the reader would have a better overview of the results of the different scenarios with and without CO without having to move from page to page to compare values. E.g. some information in Table 11 is already presented in Table 9.

L. 451-453: Not clear: were these calculations for the retention basin done within this study or do you refer to this as an example/comparison? If this was done as part of this study, I would have liked to see more information on the calculations.

L. 489-490: Could you add a sentence on how the presence of a CO reduces costs related to emergency services?

---

## Referee Comment (RC2) · Anonymous Referee #2 · 19 Aug 2020

General comments:

This publication by Ferri et al. discusses a cost-benefit analysis for citizen observatories based on a specific catchment in Italy. The content is relevant and will be a valuable addition to citizen science research. One of the current limitations of this paper is the lack of a broader context and the limited discussion. Questions that would be interesting to see addressed in the discussion section are: Why is it mostly the "social dimension of vulnerability" (L 354-355) that changes the calculations and not the additional data obtained through the CO? Is this additional data not helping to further

improve the hydraulic model? What aspect of the "social dimension of vulnerability" do the authors contribute most of the reduction in costs to? This is slightly hinted at in the methods (L 357-362), but as one of the main messages in this paper this should be addressed more extensively in the discussion. A full discussion of the results and the broader context of the study would make the value of the publication clearer to the reader.

Specific comments:

- Overall there are too many abbreviations (e.g. L 214, 258, table 6, L 369, L 379). I was not able to find the definition of the abbreviation "EWS" (table 6).

- L 55-59: Not all of the cited literature actually refers to a CO and the description of at least some of the stated studies is not accurate.

- L97-100: How often do these observations get made and how many were collected in total? It would be very informative to include a photograph of such a "staff gauge with a QR code".

- L 104: Did the volunteers operate the physical sensors? Or was this done by someone else?

- L 197 / table 1: It is not clear to me which of these data inputs are derived from citizen scientists and which are implemented anyway. Please make this distinction clearer so that the added value is more obvious.

- L 425: It would be helpful to add a range to this value, so as to show the associated uncertainty.

- L 430-432: Why do you think there is a difference, i.e. why is R3 and R4 reduced, but R1 and R2 increased? Add this to the discussion. Also table 9 does not show any areas, just damage, so the reference here probably refers to table 8?

- L 469: You mention that this method can be transferred to different catchments. It

would be interesting to read your thoughts on what type of catchments this would be suitable for, e.g. what catchment scale.

Technical corrections:

- L 92: 7th (th in superscript)

―――――――――――――――――

---

## Author Comment (AC1) · 3 Sep 2020

**Response to RC1**
We would like to thank the reviewer for their comments. These appear in bold typeface; our responses follow below each comment.

**General comments**
**Ferri et al. assess the flood risk and related costs in the Brenta-Bacchiglione catchment in Italy to evaluate the contribution of the establishment of a citizen observatory to flood risk mitigation. The authors also use this case study to demonstrate the validity of cost-benefit analysis to assess the value of citizen observatories in flood risk mitigation. As citizen science is a fairly 'hot topic' in hydrology at the moment, I think this is a timely study, providing a relevant tool that can be applied in flood risk management. The manuscript is well-written and fits well within the scope of HESS.**

Response: Thank you for these positive comments.

**I do, however, have a few questions and comments that I would like the authors to address. In the introduction you mention that several studies identified that the link of COs to authorities and policy does not necessarily lead to increased participation or improved participation. Yet, in the cost-benefit analysis with CO, you assume a positive impact of the CO on numerous social vulnerability indicators based only on the outcomes of the pilot study. I understand that the focus of this manuscript is to demonstrate the use of a cost-benefit analysis in this context, but it would nevertheless be interesting to discuss how citizen science or CO projects in other regions affected these social vulnerability indicators. This would also put the results of this study in a broader context, which is currently missing.**

Response: Based on the positive outcome of the pilot, you are correct in asserting that the CO is assumed to have positive impacts on the social vulnerability indicators. To provide a broader context, in the revised paper, we will cite the work of Bremer et al. (2019), who found that citizen science has had a high impact on adaptive capacity in a case study in Bangladesh in terms of individual awareness and understanding of local rainfall, learning that they applied in adaptive practices at work and at home as well as local leadership. Other indicators of relevance for social vulnerability refer to social capital (trust, sharing experience and formal/informal interactions) for which improvements were also measurable. However, impacts on policy were lower. So CO impacts do not necessarily (have to) work (only) via formal policy mechanisms. Both coping and adaptive capacity have individual, community as well as policy dimensions, not all of which are impacted in parallel nor to the same degree. Moreover, adaptive capacities are context-specific. We will modify the introduction to include more of this literature while returning to this in the discussion.

**This brings me to another point. From the methods, results and discussion I got the impression that most of the benefit of the citizen observatory came from the increased awareness and participation rather than just data provisioning. In the introduction, the role of citizen science and COs in data collection is actually highlighted. Also in Section 2.3, where the CO in the Brenta-Bacchiglione catchment is described, the role of 'experts' and citizens seems to focus on data acquisition, whereas the impact on estimating other flood risk drivers has not been explained as much. If you could elaborate on how the CO contributes to these aspects, this would support the (rather many) assumptions made within this study.**

Response: The increased awareness/participation and data provisioning are closely related, i.e., the app provides information about flood risk to citizens while at the same time asking for inputs/participation that can be used to feed the model and/or, in real-time, to provide information to help emergency

response. The concept of a CO is built on the idea of two-way communication between the citizens/experts and the local authorities. We can make this point clearer in the discussion to emphasize this aspect of the CO.

Regarding the impact on estimating other flood risk drivers, at present, the impact of citizens is not evaluated in the hazard component as the inputs from citizens would be used in real-time rather than the baseline modelling that was done to establish the areas flooded, the height and the flow velocity under three different flood return periods. Instead, this is incorporated into the Early Warning System (EWS) component of social vulnerability through improvements in the reliability, lead time and information content of the EWS (Figure S2 in the Supplementary Information) as well as components of Adaptive Capacity (Hazard and risk information updating and Citizen involvement - Figure S3 in the Supplementary Information). Similarly, there is currently no impact of citizens/experts on exposure or physical vulnerability as this analysis is based on land use categories rather than individual buildings, where in the latter it might be possible to capture small changes done at the household, building or feature level. However, this is not part of the current methodology. We state in the paper that hazard, exposure and physical vulnerability are not impacted by the implementation of the CO but we could make this point clearer at key points in the paper if required.

**In addition, it would be interesting to discuss at some point in the manuscript how the additional data (especially water levels) could contribute to improved hazard evaluation in your case study.**

Response: Data collected by citizens, although characterized by being asynchronous and, at times, inaccurate, can still complement traditional networks that are made up of a few highly accurate, static sensors, and hence, can improve the accuracy of the flood forecasts. For this reason, improvements to the monitoring technology have led to the spread of low-cost sensors to measure hydrological variables, such as water level, in a more distributed way. The main advantage of using this type of sensor (i.e., "social sensors") is that they can be used not only by technicians but also by any citizen. Moreover, due to their reduced cost and the voluntary labour by the citizens, they result in a more spatially distributed coverage.

In the Brenta-Bacchiglione catchment, crowdsourced observations of water level are assimilated into the hydrological model by means of rating curves assessed for the specific river location, and directly into the hydraulic model. Examples of studies we did include the following: (i) Mazzoleni et al. (2017) assessed the improvement of the flood forecasting accuracy obtained by integrating physical and social sensors distributed within the Brenta-Bacchiglione basin; and (ii) Mazzoleni et al. (2018) demonstrated that the assimilation of crowdsourced observations located at upstream points of the Bacchiglione catchment ensure high model performance for high lead times, whereas observations at the outlet of the catchments provide good results for short lead times. We can elaborate on this in the paper or in the Supplementary Material.

Mazzoleni, M., Verlaan, M., Alfonso, L., Monego, M., Norbiato, D., Ferri, M. and Solomatine, D. P.: Can assimilation of crowdsourced data in hydrological modelling improve flood prediction?, Hydrol. Earth Syst. Sci., 21(2), 839–861, doi:10.5194/hess-21-839-2017, 2017.
Mazzoleni, M., Cortes Arevalo, V. J., Wehn, U., Alfonso, L., Norbiato, D., Monego, M., Ferri, M. and Solomatine, D. P.: Exploring the influence of citizen involvement on the assimilation of crowdsourced observations: a modelling study based on the 2013 flood event in the Bacchiglione catchment (Italy), Hydrol. Earth Syst. Sci., 22(1), 391–416, doi:10.5194/hess-22-391-2018, 2018.

**More specific comments and requests for further clarification on certain points in the manuscript are provided below.**

**Specific comments L. 19-20: I would use citizen observatories in this sentence as well, since your manuscript evaluates how these can contribute to risk reduction.**

Response: We have added citizen observatories to this sentence. It now reads as follows:

Thus, linking citizen science and citizen observatories with hydrological modelling to raise awareness of flood hazards and to facilitate two-way communication between citizens and local authorities has great potential in reducing future flood risk in the Brenta-Bacchiglione catchment.

**Figure 1: Please add legend to map and clearly indicate the boundaries of the Brenta-Bacchiglione catchment.**

Response: Below is a figure clearly showing the boundaries of the Brenta-Bacchiglione catchment and the river network. In the revised version of the paper, we will add a legend to Figure 1 and clearly show the boundaries of the catchment.

[Figure]

Figure showing the location of the Brenta-Bacchiglione catchment (red line) and its main rivers (in blue).

**L. 143-145: What is the sustainability of such an arrangement, whereby the technicians get paid for each trip, once the project ends? Where would the funds come from?**

Table S2 contains the costs of the components of the Citizen Observatory (CO) for Flood Risk Management where we have foreseen a financial safety net of 5 years to develop and operate all components of the CO (for developing the technology, maintenance, education campaigns, etc.). After this period, we will evaluate the results and pursue the opportunity to fund the initiative further.

**L. 151-159: Could you be more specific on the kind of observations that citizens can contribute? I would imagine these are less 'complicated' than the contributions of the trained volunteers and technicians.**

**Furthermore – as mentioned in the general comments – how will citizen engage further in risk reduction such that flood risks can be reduced?**

Citizens can easily send and share reports regarding measured hydrological quantities, for example: the water level of a river at a section equipped with a hydrometric measuring rod and QR code or the level of the snowpack where a snow gauge equipped with a QR code has been installed. They can also send reports about the presence of flooded areas indicating the water height. They can also send 'smart' measurements, which are simplified measurements of some hydrological variables (e.g., the amount of rain, the forest cover of the banks, weather conditions, etc.). Instead of entering a value, citizens can choose from a series of photos/videos that represent the phenomenon, and which are associated with ranges of values (e.g., for the intensity of the rain: drizzle, moderate, heavy, very heavy). The citizen chooses the photo most similar to what they see when they send the report.

In exchange, citizens can receive flood-related information (e.g., weather and river level forecasts, notifications from the authority concerning the declaration of a state of alert or its cessation, specific communications to citizens present in a specific area of interest/danger in a specific period of time (based on a geolocation function). This two-way communication can help to reduce flood risks.

Technicians, as any citizen, can send alerts or observations but they can also provide technical data (e.g., the status of the embankments, vegetation cover, water levels, clearance below the bridge) that are considered as trusted (and already validated) because of their expertise.

**L. 185: Supplementary tables: Please change the numbers of the supplementary tables in such a way that they are numbered according to their appearance in the text.**

Response: Thank you for spotting this error. We have now made these changes, renumbering the supplementary tables in the order that they appear in the text and reordering them in the supplementary material.

**L. 223-227: The section on EE and EECH is quite unclear and could do with some more detail, as I cannot really follow what has been done based on the current description.**

The EU Flood Directive and Italian law requires a description of the type and spatial distribution of the economic activities in the flooded area in order to provide an assessment of the potential negative consequences for the different activities. The relative exposure of economic activities ($E_E$) was used in the methodology as the physical exposure, expressed by the restoration costs, the costs due to missed production and service losses. The values used to represent this economic factor are reported in Table 3.

Similarly, to define the exposure values for the environmental and cultural heritage component ($E_{ECH}$), we proceed by assigning relative values to different land use categories, taking into account the potential modifications that can occur during an adverse event to the various environmental and cultural features contained within these different land use categories. Those values are also presented in Table 3.

**Table 3: Are these relative values based on the data explained in the previous paragraphs? This seems straightforward for EP, but it is not clear how EE and EECH translate from costs to a relative value. And how have these values been tested and shown to be valid?**

Yes, these relative values are based on the data explained in the previous paragraph. These values have been derived by the Provincia Autonoma di Trento (2006) from decades of experience with understanding exposure related to flood risk. In Tables A and B, we have provided further information on the relation between the costs and the relative value.

Table A: The logic used to assign relative values to the categories of $E_E$

| Category | $E_E$ | Description |
|---|---|---|
| Residential | 1 | The costs of restoring the buildings, those of the assets kept in them, and those relating to the accommodation of people during the restoration period are high. |
| Industrial | 1 | The costs of restoring buildings, those of the equipment present, and those of non-production during the period of inactivity are high. |
| Specialized agricultural | 0.3 - 1 | Average costs for the restoration of crops and the average costs for the lack of harvest. Not being able to differentiate between different types of crops, it was still necessary to differentiate between crops of very low value (e.g., corn: the harvest has a very low value and recovery times are short) and others of high value (e.g., vineyards: the harvest can be very valuable and recovery times can be very long). |
| Woods, meadows, pastures, cemeteries, urban parks | 0.3 | Low restoration costs and low costs related to the lack of timber production |
| Unproductive | 0.1 | Very low economic value |
| Ski areas | 0.3 | The costs of restoring the infrastructure and those related to the failure to use the service during the restoration period are low |
| Roads of primary importance | 1 | The costs of restoring the infrastructure and costs related to the failure to use the service during the restoration period are high |
| Landfill, Waste treatment plants, Mining areas, Purifiers | 0.5 | Average costs for the restoration of the various types of works and low costs regarding failure to carry out the service (e.g., in the event of a disaster, the wastewater ends up in the river and the waste is disposed of elsewhere) |
| Areas of historical, cultural and archaeological importance | 1 | High costs associated with the restoration of buildings |

Table B: The logic used to assign the relative values to categories of $E_{ECH}$

| Category | $E_{ECH}$ | Description |
|---|---|---|
| Residential | 1 | High historical, cultural and environmental value |
| Industrial | 0.3 - 1 | Medium-high environmental value since the presence of industry is negative for the environment and therefore its disappearance would improve the environmental quality, but its damage could be negative as it could generate pollution. |
| Specialized agricultural | 0.7 | Medium environmental and cultural value |
| Woods, meadows, pastures, cemeteries, urban parks | 0.7 | Medium environmental and cultural value |
| Unproductive | 0.3 | If the unproductive land is affected by a hydrogeological event, the quality of the environment does not change as the environment is not substantially changed |
| Ski areas | 0.3 | Medium environmental value since the presence of the ski area is in itself negative for the environment and therefore its disappearance would improve the environmental quality, but it is instead positive from a recreational point of view |
| Roads of primary importance | 0.2 | The presence of primary roads is in itself very negative from a landscape point of view and therefore its disappearance would improve the environmental quality |
| Landfill, Waste treatment plants, Mining areas, Purifiers | 1 | High environmental value due to the pollution that would be generated in the event of service interruption and therefore a serious deterioration in environmental quality |
| Areas on which plants are installed as per Annex I of Legislative Decree 18 February 2005, n. 59 | 1 | High environmental value since the damage could be negative as it could generate pollution |
| Areas of historical, cultural and archaeological importance | 1 | High historical, cultural and environmental value |
| Environmental goods | 1 | High historical, cultural and environmental value |

**Figure 3: I would place this figure in Section (ii), where you explain how the weights and values have been obtained. How did you define the indicators? Is this based on existing literature? And could you provide more information on the stakeholder engagement to identify weights and values? Which stakeholders were engaged and how were the weights and values determined? Average of what different stakeholders provided? Or did certain stakeholder provide info for certain indicators only?**

We agree with the reviewer and we have moved Figure 3 to Section (ii). The indicators in Figure 3 are based on existing literature (Mojtahed et al., 2013). The weights given to each indicator were derived from an expert consultation process. The principal aim of this procedure was to assign a value between 0 and 1 to people's vulnerability, considering the relative weight of each indicator. The stakeholders engaged were the members of the Technical Committee of the water basin authority made up of technical representatives of the regional and provincial administrations belonging to the Eastern Alps District, as well as experts from the professional and academic sectors (i.e., around 20 people). The process to identify the weights started with several discussions, the results of which were interpreted and translated into values/weights by AAWA, who then re-proposed to the experts, obtaining their consensus.

Mojtahed, V., Giupponi, C., Biscaro, C., Gain, A. K. and Balbi, S.: Integrated Assessment of Natural Hazards and Climate Change Adaptation: The SERRA Methodology. Università Cà Foscari of Venice, Dept of Economics Research Paper Series No. 07/WP/2013, 2013.

**Table 4: Include the references to your data sources in the table (either in the caption or as footnote) if the values are directly taken from the studies you mention in L. 252-254.**

Response: We have added the following source to the Table 4 caption:

Source: ISPRA (2012), with reference to DEFRA and UK Environment Agency (2006)

**Figure 4 and similar figures: Are only the values 0.25, 0.75 and 1.0 included in the analysis? If so, why particularly these values and not values in between?**

These values are the ones proposed by the ISPRA guidelines (ISPRA, 2012).

**L. 266-268: How were the value functions for the different indicators defined? Was this also part of the stakeholder consultation?**

Similarly to what was done to identify the weights in Figure 3, AAWA formulated an internal study for the definition of the value functions for the different indicators, which were then proposed and discussed with the members of a Technical Committee (see response above), obtaining their consensus.

**L. 284-288: How were content and reliability quantified? Simply assigning it to one of the categories shown in Figure S2 could be quite subjective.**

Content and reliability were assigned to one of the categories shown in Figure S2 based on the following assumptions:
- Reliability: The Early Warning System (EWS) reliability increases to very high due to the involvement of trained citizens who provide information and sensor readings that are used to validate and feed the hydrological/hydraulic model (i.e., the data assimilation module). The assumption was made based on the results obtained in Mazzoleni et al. (2017, 2018) and by

considering a hypothetical situation in which a widely distributed crowdsourcing data acquisition process is in place due to the expected high level of citizen engagement.

- Content: With the CO, we will be obtaining very detailed information further enriched by citizen reports (including reports from waterways that were not previously equipped with measuring instruments) and by a monitoring network that will be equipped with a further eight thermo-pluviometric stations, 12 hydrometric stations (equipped with a double transmission system), and 58 hydrometric and six snow measuring rods. The forecasted water level is available at every section of the Brenta-Bacchiglione River system. Overall the content is enhanced through the implementation of the CO.

**L. 299-302: More information on this is required. From what is provided, it is difficult to understand how to interpret Figure S3d. What does no category mean? That no citizens are involved (which would correspond to zero citizen involvement and thus increased vulnerability)?**

Response: 'No category' has been changed in Figure S3d to 'No involvement', i.e. this corresponds to zero citizen involvement and thus increased vulnerability. We have modified Figure S3d as shown below.

[Figure]

**L. 318-319: In the caption of Figure S3, it says these values are for network infrastructure. Does this only apply to roads? If so, please change accordingly.**

Response: Here we assume that you are referring to Figure S5. Yes, the network infrastructure applies only to roads, so we have changed the figure caption to read: Vulnerability values of the road infrastructure as a function of water height (h) and flow velocity (v).

**L. 356-358: If social vulnerability decreases to 0.63, what was its original value?**

The original value was 0.9.

**Tables 8-11: If you combine the tables, the reader would have a better overview of the results of the different scenarios with and without CO without having to move from page to page to compare values. E.g. some information in Table 11 is already presented in Table 9.**

Response: Tables 8 and 10 have been combined to more clearly show the areas by risk class before and after implementation of a CO into a new Table 8:

Table 8: The risk classes for each return period in terms of area flooded (km$^2$) before and after implementation of the CO.

| Risk Class | Before implementation of the CO | | | After implementation of the CO | | |
|---|---|---|---|---|---|---|
| | 30 year return period | 100 year return period | 300 year return period | 30 year return period | 100 year return period | 300 year return period |
| Low (R1) | 160.29 | 254.29 | 318.80 | 170.96 | 268.68 | 337.78 |
| Medium (R2) | 137.26 | 191.89 | 262.03 | 168.99 | 235.18 | 322.41 |
| High (R3) | 56.70 | 79.23 | 110.29 | 18.19 | 27.19 | 40.04 |
| Very High (R4) | 3.92 | 5.73 | 9.23 | 0.03 | 0.09 | 0.12 |
| Total | 358.17 | 531.14 | 700.35 | 358.17 | 531.14 | 700.35 |

Table 9 has been removed as the data already appear in Table 11, where Table 11 has subsequently been renumbered to Table 9. This has also required some reorganization of the text, but we agree that combining the results for before and after the CO implementation are clearer for the reader.

**L. 451-453: Not clear: were these calculations for the retention basin done within this study or do you refer to this as an example/comparison? If this was done as part of this study, I would have liked to see more information on the calculations.**

Response: We refer to this only as an example since the retention basin calculations were done as part of another study independent of this CO.

**L. 489-490: Could you add a sentence on how the presence of a CO reduces costs related to emergency services?**

Although the people involved in the emergency services are the same, they are employed in a much more efficient way as a result of the technology developed within the CO, which allows for better management of the teams responding to the event and the assignment of tasks based on an operator's location. For example, the authorities can assign critical tasks to rescue teams and produce reports on progress, they can monitor the movements of the teams, and they can assign new tasks to teams once a job is finished based on the proximity job principle. Furthermore, civil protection plans can be updated more frequently, which also draw upon more active citizen participation in reporting risk situations in their surroundings.

---

## Author Comment (AC2) · 3 Sep 2020

**Response to RC2**

We would like to thank the reviewer for their comments. These appear in bold typeface; our responses follow below each comment.

**General comments**

**This publication by Ferri et al. discusses a cost-benefit analysis for citizen observatories based on a specific catchment in Italy. The content is relevant and will be a valuable addition to citizen science research. One of the current limitations of this paper is the lack of a broader context and the limited discussion. Questions that would be interesting to see addressed in the discussion section are: Why is it mostly the "social dimension of vulnerability" (L 354-355) that changes the calculations and not the additional data obtained through the CO? Is this additional data not helping to further improve the hydraulic model? What aspect of the "social dimension of vulnerability" do the authors contribute most of the reduction in costs to? This is slightly hinted at in the methods (L 357-362), but as one of the main messages in this paper this should be addressed more extensively in the discussion. A full discussion of the results and the broader context of the study would make the value of the publication clearer to the reader.**

Response: We thank the reviewer for their positive comments regarding relevance and a valuable addition to citizen science research. We agree that the discussion could be improved to provide a broader context. For example, the additional data obtained through the CO are indirectly part of the improvements to social vulnerability rather than the hazard component that involves the hydrological-hydraulic model as one might be expecting. This is because the hazard modelling is done as a baseline while the data collected by the citizens/expert volunteers will be in near-real time so this will affect the early warning system, the involvement and hence awareness of citizens, and the updating of the hazard and risk information, which are all part of social vulnerability.

One of the aspects that contributes most to a reduction in costs is related to the **emergency services.** Although the people involved in the emergency services are the same, they are employed in a much more efficient way as a result of the technology developed within the CO, which allows for better management of the teams responding to the event and the assignment of tasks based on an operator's location. For example, the authorities, in response to real-time reports from citizens and based on reliable model forecasts, can assign critical tasks to rescue teams, monitor the movements of the teams and assign new tasks to teams once a job is finished based on the proximity job principle. Secondly, among the factors of social vulnerability, the involvement of citizens contributes to changing those behaviours that are the main causes of death and/or serious economic damage; an example would be actions such as trying to save your car during a flood event or trying to rescue your belongings from a flooded basement, which would be stopped or reduced as a result of the CO.

We will add this type of additional context to the discussion to address the limitation pointed out by the reviewer.

**Specific comments:**

**Overall there are too many abbreviations (e.g. L 214, 258, table 6, L 369, L 379). I was not able to find the definition of the abbreviation "EWS" (table 6).**

Response: Regarding L214, this is a blank line in our version of the paper unless you are referring to L215, which defines one of the variables in equation 2? Regarding L258, is the reviewer referring to FHR or Vp as an acronym? If it is FHR, this was defined further up in the text and used in equation 3. If Vp, then we

have now added this to the caption of Figure 4. In Table 6, we have changed the acronym EWS to read Early Warning System as this was not defined previously in the paper so thank you for pointing this out. On L369, CLC refers to Corine Land Cover, which we defined in L192/193. L379 defines variables that are then used in equations 5 and 6 and further in the text. However, we take the reviewer's point that there are many acronyms and variables defined and used throughout the paper. We would be happy to create a list of acronyms if this would help or remove some of the acronyms and replace them with the full text when they are not used very often in the paper.

**L 55-59: Not all of the cited literature actually refers to a CO and the description of at least some of the stated studies is not accurate.**

Response: We have checked the referenced literature and deleted those references that do not refer to COs (Etter et al., 2018; Mazzonleni et al., 2017; Butaert et al., 2014) but to other forms of citizen science. Moreover, we have attributed the statement regarding the link of COs with authorities and policy more specifically as follows:

*"Specifically, Wehn et al. (2015) found that the characteristic links of COs to authorities and policy do not automatically translate into higher levels of participation in flood risk management, nor that communication between stakeholders improves; rather, changes towards fundamentally more involved citizen roles with higher impact in flood risk management can take years to evolve."*

**L97-100: How often do these observations get made and how many were collected in total? It would be very informative to include a photograph of such a "staff gauge with a QR code".**

During the WeSenseIt project, more than two hundred people were recruited for practical activities and were trained to use the WeSenseIt technologies. The data collected took place during the evaluation exercises organized to test the technology and to collect feedback for further development and to make improvements. During these events, the response of the volunteers was enthusiastic as well as their participation in sending environmental reports and information. Examples of photographs are provided below, which we could add to the paper.

[Figure]

**L 104: Did the volunteers operate the physical sensors? Or was this done by someone else?**

Response: No, the physical sensors are operated by AAWA in collaboration with the Regional Department for Soil Protection, the Environmental Agency, the Civil Protection Agency and their related professionals.

**L 197 / table 1: It is not clear to me which of these data inputs are derived from citizen scientists and which are implemented anyway. Please make this distinction clearer so that the added value is more obvious.**

Response: None of the data in Table 1 (version posted online) are derived from citizen scientists. We have now added the input from citizen scientists to the Flood Vulnerability component as follows:

Table 1: Input data used to calculate risk.

| Component of risk | Data | Source |
|---|---|---|
| Flood Hazard | Same as before | Same as before |
| Flood Exposure | Same as before | Same as before |
| Flood Vulnerability (Susceptibility) | Vegetation cover | Corine Land Cover 2006 |
| | Soil type | Corine Land Cover 2006 |
| | Water height from simple gauges equipped with QR codes, which are read by technicians and citizens as well as photographs and other flood-relevant information collected via an app | Collected by AAWA |

These data are used indirectly in the calculation of social vulnerability, i.e., the Early Warning System (EWS) component through improvements in the reliability, lead time and information content of the EWS (Figure S2 in the Supplementary Information) as well as two components of Adaptive Capacity (Hazard and risk information updating and Citizen involvement - Figure S3 in the Supplementary Information).

**L 425: It would be helpful to add a range to this value, so as to show the associated uncertainty.**

Although we understand the point of the reviewer, we do not have a range around the expected average annual damage because it is already based on three probabilistic flood scenarios, so this damage value is an average across these scenarios. We recognize that there are multiple assumptions and uncertainties in this methodology, but we have not quantified them as such. Since the exercise is currently a hypothetical one but it provided sufficient evidence for funding of the CO for five years, going back and doing a full uncertainty analysis would not bring any further value for the operational running of the CO. Instead, once the CO is operational, it will then be more interesting for us to verify the results from the cost-benefit analysis.

**L 430-432: Why do you think there is a difference, i.e. why is R3 and R4 reduced, but R1 and R2 increased? Add this to the discussion.**

Response: This occurs because the total area affected by the flood hazard is the same before and after implementation of a CO. What changes is the distribution between risk classes, i.e., R3 and R4 are reduced, which means that the areas at risk in classes R1 and R2 will increase. We will add this to the discussion regarding the implications of this finding.

**Also table 9 does not show any areas, just damage, so the reference here probably refers to table 8?**

Response: Thank you for pointing out this error. We have corrected this but in the process of responding to reviewer 1, we have also combined tables, i.e., Table 8 and 10 are now combined (risk before and after the implementation of a CO in terms of area) and Table 9 and 11 (damage in euro amounts before and after the implementation of a CO). The text has also been updated accordingly.

**L 469: You mention that this method can be transferred to different catchments. It would be interesting to read your thoughts on what type of catchments this would be suitable for, e.g. what catchment scale.**

Once activated in the Brenta-Bacchiglione, the CO will also be extended to the other basins of the hydrographic district of the Eastern Alps, which are similar in size and hydrological characteristics. These are complex hydrographic basins with very variable regimes, from rapid response/torrential rainfall events of the alpine territories to the alluvial plain, which is composed of mountain and lowland river networks, artificial networks of reclamation and natural and/or artificial reservoirs. In general, we suggest applying this methodology to catchments larger than 100 km$^2$. For catchments of this size and greater, we would have basin compatible response times, and hence, it would make sense to use model forecasts.

Another point to note is that for the application of the CO methodology, it is necessary that the population residing in the basin can be easily reached through such an initiative, and that they are familiar with, and are able to access, the technology (i.e., via a tablet, PC, smartphone).

**Technical corrections:**
**L 92: 7th (th in superscript)**

Response: Thank you for spotting this error. We have now corrected this.

---

## Author Response (AR1)

**Response to RC1**
We would like to thank the reviewer for their comments. These appear in bold typeface; our responses follow below each comment.

**General comments**
**Ferri et al. assess the flood risk and related costs in the Brenta-Bacchiglione catchment in Italy to evaluate the contribution of the establishment of a citizen observatory to flood risk mitigation. The authors also use this case study to demonstrate the validity of cost-benefit analysis to assess the value of citizen observatories in flood risk mitigation. As citizen science is a fairly 'hot topic' in hydrology at the moment, I think this is a timely study, providing a relevant tool that can be applied in flood risk management. The manuscript is well-written and fits well within the scope of HESS.**

Response: Thank you for these positive comments.

**I do, however, have a few questions and comments that I would like the authors to address. In the introduction you mention that several studies identified that the link of COs to authorities and policy does not necessarily lead to increased participation or improved participation. Yet, in the cost-benefit analysis with CO, you assume a positive impact of the CO on numerous social vulnerability indicators based only on the outcomes of the pilot study. I understand that the focus of this manuscript is to demonstrate the use of a cost-benefit analysis in this context, but it would nevertheless be interesting to discuss how citizen science or CO projects in other regions affected these social vulnerability indicators. This would also put the results of this study in a broader context, which is currently missing.**

Response: Based on the positive outcome of the pilot, you are correct in asserting that the CO is assumed to have positive impacts on the social vulnerability indicators. To provide a broader context, we have now included reference to the work of Bremer et al. (2019), who found in their case study in Bangladesh that citizen science has had a high impact on adaptive capacity in terms of individual awareness and understanding of local rainfall, learning that they applied in adaptive practices at work and at home, as well as local leadership. Other relevant indicators of social vulnerability refer to social capital (trust, sharing experience and formal/informal interactions) for which improvements were also measurable. However, impacts on policy were lower. So CO impacts do not necessarily (have to) materialise (only) via formal policy mechanisms. Both coping and adaptive capacity have individual, community as well as policy dimensions, not all of which are impacted in parallel nor to the same degree. Moreover, adaptive capacities are context specific.

**This brings me to another point. From the methods, results and discussion I got the impression that most of the benefit of the citizen observatory came from the increased awareness and participation rather than just data provisioning. In the introduction, the role of citizen science and COs in data collection is actually highlighted. Also in Section 2.3, where the CO in the Brenta-Bacchiglione catchment is described, the role of 'experts' and citizens seems to focus on data acquisition, whereas the impact on estimating other flood risk drivers has not been explained as much. If you could elaborate on how the CO contributes to these aspects, this would support the (rather many) assumptions made within this study.**

Response: The increased awareness/participation and data provisioning are closely related, i.e., the app provides information about flood risk to citizens while at the same time asking for inputs/participation that can be used to feed the model and/or, in real-time, to provide information to help emergency response. The concept of a CO is built on the idea of two-way communication between the

citizens/experts and the local authorities. We can make this point clearer in the discussion to emphasize this aspect of the CO.

Regarding the impact on estimating other flood risk drivers, at present, the impact of citizens is not evaluated in the hazard component as the inputs from citizens would be used in real-time rather than the baseline modelling that was done to establish the areas flooded, the height and the flow velocity under three different flood return periods. Instead, this is incorporated into the Early Warning System (EWS) component of social vulnerability through improvements in the reliability, lead time and information content of the EWS (Figure S2 in the Supplementary Information) as well as components of Adaptive Capacity (Hazard and risk information updating and Citizen involvement - Figure S3 in the Supplementary Information). Similarly, there is currently no impact of citizens/experts on exposure or physical vulnerability as this analysis is based on land use categories rather than individual buildings, where in the latter it might be possible to capture small changes done at the household, building or feature level. However, this is not part of the current methodology. We state in the paper that hazard, exposure and physical vulnerability are not impacted by the implementation of the CO. However, we have added a paragraph to the Discussion and Conclusion section that highlights these points.

**In addition, it would be interesting to discuss at some point in the manuscript how the additional data (especially water levels) could contribute to improved hazard evaluation in your case study.**

Response: Data collected by citizens, although characterized by being asynchronous and, at times, inaccurate, can still complement traditional networks that are made up of a few highly accurate, static sensors, and hence, can improve the accuracy of the flood forecasts. For this reason, improvements to the monitoring technology have led to the spread of low-cost sensors to measure hydrological variables, such as water level, in a more distributed way. The main advantage of using this type of sensor (i.e., "social sensors") is that they can be used not only by technicians but also by any citizen. Moreover, due to their reduced cost and the voluntary labour by the citizens, they result in a more spatially distributed coverage.

In the Brenta-Bacchiglione catchment, crowdsourced observations of water level are assimilated into the hydrological model by means of rating curves assessed for the specific river location, and directly into the hydraulic model. Examples of studies we did include the following: (i) Mazzoleni et al. (2017) assessed the improvement of the flood forecasting accuracy obtained by integrating physical and social sensors distributed within the Brenta-Bacchiglione basin; and (ii) Mazzoleni et al. (2018) demonstrated that the assimilation of crowdsourced observations located at upstream points of the Bacchiglione catchment ensure high model performance for high lead times, whereas observations at the outlet of the catchments provide good results for short lead times.

We have added some of this explanation to section 3.2.1 on Flood Hazard Mapping.

Mazzoleni, M., Verlaan, M., Alfonso, L., Monego, M., Norbiato, D., Ferri, M. and Solomatine, D. P.: Can assimilation of crowdsourced data in hydrological modelling improve flood prediction?, Hydrol. Earth Syst. Sci., 21(2), 839–861, doi:10.5194/hess-21-839-2017, 2017.

Mazzoleni, M., Cortes Arevalo, V. J., Wehn, U., Alfonso, L., Norbiato, D., Monego, M., Ferri, M. and Solomatine, D. P.: Exploring the influence of citizen involvement on the assimilation of crowdsourced observations: a modelling study based on the 2013 flood event in the Bacchiglione catchment (Italy), Hydrol. Earth Syst. Sci., 22(1), 391–416, doi:10.5194/hess-22-391-2018, 2018.

**More specific comments and requests for further clarification on certain points in the manuscript are provided below.**

**Specific comments L. 19-20: I would use citizen observatories in this sentence as well, since your manuscript evaluates how these can contribute to risk reduction.**

Response: We have added citizen observatories to this sentence. It now reads as follows:

Thus, linking citizen science and citizen observatories with hydrological modelling to raise awareness of flood hazards and to facilitate two-way communication between citizens and local authorities has great potential in reducing future flood risk in the Brenta-Bacchiglione catchment.

**Figure 1: Please add legend to map and clearly indicate the boundaries of the Brenta-Bacchiglione catchment.**

Response: We have added the boundaries of the Brenta-Bacchiglione catchment and the river network to Figure 1 (which is now Figure 2). We have also added a legend.

**L. 143-145: What is the sustainability of such an arrangement, whereby the technicians get paid for each trip, once the project ends? Where would the funds come from?**

Table S6 contains the costs of the components of the Citizen Observatory (CO) for Flood Risk Management where we have foreseen a financial safety net of 5 years to develop and operate all components of the CO (for developing the technology, maintenance, education campaigns, etc.). After this period, we will evaluate the results and pursue the opportunity to fund the initiative further. We added a line to this effect where Table S6 is referred to in the results.

**L. 151-159: Could you be more specific on the kind of observations that citizens can contribute? I would imagine these are less 'complicated' than the contributions of the trained volunteers and technicians. Furthermore – as mentioned in the general comments – how will citizen engage further in risk reduction such that flood risks can be reduced?**

Citizens can easily send and share reports regarding measured hydrological quantities, for example: the water level of a river at a section equipped with a hydrometric measuring rod and QR code or the level of the snowpack where a snow gauge equipped with a QR code has been installed. They can also send reports about the presence of flooded areas indicating the water height. They can also send 'smart' measurements, which are simplified measurements of some hydrological variables (e.g., the amount of rain, the forest cover of the banks, weather conditions, etc.). Instead of entering a value, citizens can choose from a series of photos/videos that represent the phenomenon, and which are associated with ranges of values (e.g., for the intensity of the rain: drizzle, moderate, heavy, very heavy). The citizen chooses the photo most similar to what they see when they send the report.

In exchange, citizens can receive flood-related information (e.g., weather and river level forecasts, notifications from the authority concerning the declaration of a state of alert or its cessation, specific communications to citizens present in a specific area of interest/danger in a specific period of time (based on a geolocation function). This two-way communication can help to reduce flood risks.

We have added parts of this text above to section 2.3 to provide additional clarification.

Note that technicians, as any citizen, can send alerts or observations but they can also provide technical data (e.g., the status of the embankments, vegetation cover, water levels, clearance below the bridge) that are considered as trusted (and already validated) because of their expertise.

**L. 185: Supplementary tables: Please change the numbers of the supplementary tables in such a way that they are numbered according to their appearance in the text.**

Response: Thank you for spotting this error. We have now made these changes, renumbering the supplementary tables in the order that they appear in the text and reordering them in the supplementary material. We marked these changes in yellow in the supplementary material document.

**L. 223-227: The section on EE and EECH is quite unclear and could do with some more detail, as I cannot really follow what has been done based on the current description.**

The EU Flood Directive and Italian law requires a description of the type and spatial distribution of the economic activities in the flooded area in order to provide an assessment of the potential negative consequences for the different activities. The relative exposure of economic activities ($E_E$) was used in the methodology as the physical exposure, expressed by the restoration costs, the costs due to missed production and service losses. The values used to represent this economic factor are reported in Table 3.

Similarly, to define the exposure values for the environmental and cultural heritage component ($E_{ECH}$), we proceed by assigning relative values to different land use categories, taking into account the potential modifications that can occur during an adverse event to the various environmental and cultural features contained within these different land use categories. Those values are also presented in Table 3.

We have rewritten this section to try to make this explanation clearer. We have also provided Tables S3 and S4, which provides the reasoning for how the costs are translated into relative values. See the next response.

**Table 3: Are these relative values based on the data explained in the previous paragraphs? This seems straightforward for EP, but it is not clear how EE and EECH translate from costs to a relative value. And how have these values been tested and shown to be valid?**

The relative values are based on the data explained in the previous paragraph and have been derived by the Provincia Autonoma di Trento (2006) from decades of experience with understanding exposure related to flood risk. We have added Tables S3 and S4 to the Supplementary Material to provide further information on the relation between the costs and the relative values.

**Figure 3: I would place this figure in Section (ii), where you explain how the weights and values have been obtained. How did you define the indicators? Is this based on existing literature? And could you provide more information on the stakeholder engagement to identify weights and values? Which stakeholders were engaged and how were the weights and values determined? Average of what different stakeholders provided? Or did certain stakeholder provide info for certain indicators only?**

We agree with the reviewer and we have moved Figure 3 to Section (ii). The indicators in Figure 3 (now Figure 5) are based on existing literature (Mojtahed et al., 2013). The weights given to each indicator were derived from an expert consultation process. The principal aim of this procedure was to assign a value between 0 and 1 to people's vulnerability, considering the relative weight of each indicator. The

stakeholders engaged were the members of the Technical Committee of the water basin authority made up of technical representatives of the regional and provincial administrations belonging to the Eastern Alps District, as well as experts from the professional and academic sectors (i.e., around 20 people). The process to identify the weights started with several discussions, the results of which were interpreted and translated into values/weights by AAWA, who then re-proposed these to the experts, obtaining their consensus. We have added this to the text to provide clarification.

Mojtahed, V., Giupponi, C., Biscaro, C., Gain, A. K. and Balbi, S.: Integrated Assessment of Natural Hazards and Climate Change Adaptation: The SERRA Methodology. Università Cà Foscari of Venice, Dept of Economics Research Paper Series No. 07/WP/2013, 2013.

**Table 4: Include the references to your data sources in the table (either in the caption or as footnote) if the values are directly taken from the studies you mention in L. 252-254.**

Response: We have added the following source to the Table 4 caption: Source: ISPRA (2012), with reference to DEFRA and UK Environment Agency (2006)

**Figure 4 and similar figures: Are only the values 0.25, 0.75 and 1.0 included in the analysis? If so, why particularly these values and not values in between?**

These three values of Vp were proposed in the ISPRA (2012) guidelines. This has been added to the text. We have also added this to relevant figures in the Supplementary Material.

**L. 266-268: How were the value functions for the different indicators defined? Was this also part of the stakeholder consultation?**

Similarly to what was done to identify the weights in Figure 3 (now Figure 5), AAWA formulated an internal study for the definition of the value functions for the different indicators, which were then proposed and discussed with the members of a Technical Committee (see response above), obtaining their consensus. We added an additional line to the text to clarify this (following the explanation provided to the comment above about Figure 3.

**L. 284-288: How were content and reliability quantified? Simply assigning it to one of the categories shown in Figure S2 could be quite subjective.**

Content and reliability were assigned to one of the categories shown in Figure S2 based on the following assumptions:
- Reliability: The Early Warning System (EWS) reliability increases to very high due to the involvement of trained citizens who provide information and sensor readings that are used to validate and feed the hydrological/hydraulic model (i.e., the data assimilation module). The assumption was made based on the results obtained in Mazzoleni et al. (2017, 2018) and by considering a hypothetical situation in which a widely distributed crowdsourcing data acquisition process is in place due to the expected high level of citizen engagement.
- Content: With the CO, we will be obtaining very detailed information further enriched by citizen reports (including reports from waterways that were not previously equipped with measuring instruments) and by a monitoring network that will be equipped with a further eight thermo-pluviometric stations, 12 hydrometric stations (equipped with a double transmission system), and 58 hydrometric and six snow measuring rods. The forecasted water level is available at every

section of the Brenta-Bacchiglione River system. Overall the content is enhanced through the implementation of the CO.

These explanations were added close to Table 6, which explained the changes in social vulnerability before and after implementation of a CO on flood risk management.

**L. 299-302: More information on this is required. From what is provided, it is difficult to understand how to interpret Figure S3d. What does no category mean? That no citizens are involved (which would correspond to zero citizen involvement and thus increased vulnerability)?**

Response: 'No category' has been changed in Figure S3d to 'No involvement', i.e., this corresponds to zero citizen involvement and thus increased vulnerability. We have modified Figure S3d and updated it in the Supplementary Material. The caption is highlighted in yellow to flag this changed figure.

**L. 318-319: In the caption of Figure S3, it says these values are for network infrastructure. Does this only apply to roads? If so, please change accordingly.**

Response: Here we assume that you are referring to Figure S5. Yes, the network infrastructure applies only to roads, so we have changed the figure caption to read: Vulnerability values of the road infrastructure as a function of water height (h) and flow velocity (v).

**L. 356-358: If social vulnerability decreases to 0.63, what was its original value?**

The original value was 0.9. We have added this figure to the text.

**Tables 8-11: If you combine the tables, the reader would have a better overview of the results of the different scenarios with and without CO without having to move from page to page to compare values. E.g. some information in Table 11 is already presented in Table 9.**

Response: Tables 8 and 10 have been combined to more clearly show the areas by risk class before and after implementation of a CO into a new Table 8. Table 9 has been removed as the data already appear in Table 11, where Table 11 has subsequently been renumbered to Table 9. This has also required some reorganization of the text, but we agree that combining the results for before and after the CO implementation are clearer for the reader.

**L. 451-453: Not clear: were these calculations for the retention basin done within this study or do you refer to this as an example/comparison? If this was done as part of this study, I would have liked to see more information on the calculations.**

Response: We refer to this only as an example since the retention basin calculations were done as part of another study independent of this CO.

**L. 489-490: Could you add a sentence on how the presence of a CO reduces costs related to emergency services?**

Although the people involved in the emergency services are the same, they are employed in a much more efficient way as a result of the technology developed within the CO, which allows for better management

of the teams responding to the event and the efficient assignment of tasks based on an operator's location.

We have added the above statement to the discussion.

For further clarification, the authorities can assign critical tasks to rescue teams and produce reports on progress, they can monitor the movements of the teams, and they can assign new tasks to teams once a job is finished based on the proximity job principle. Furthermore, civil protection plans can be updated more frequently, which also draw upon more active citizen participation in reporting risk situations in their surroundings.

**Response to RC2**

We would like to thank the reviewer for their comments. These appear in bold typeface; our responses follow below each comment.

**General comments**
**This publication by Ferri et al. discusses a cost-benefit analysis for citizen observatories based on a specific catchment in Italy. The content is relevant and will be a valuable addition to citizen science research. One of the current limitations of this paper is the lack of a broader context and the limited discussion. Questions that would be interesting to see addressed in the discussion section are: Why is it mostly the "social dimension of vulnerability" (L 354-355) that changes the calculations and not the additional data obtained through the CO? Is this additional data not helping to further improve the hydraulic model? What aspect of the "social dimension of vulnerability" do the authors contribute most of the reduction in costs to? This is slightly hinted at in the methods (L 357-362), but as one of the main messages in this paper this should be addressed more extensively in the discussion. A full discussion of the results and the broader context of the study would make the value of the publication clearer to the reader.**

Response: We thank the reviewer for their positive comments regarding relevance and a valuable addition to citizen science research. The additional data obtained through the CO are indirectly part of the improvements to social vulnerability rather than the hazard component that involves the hydrological-hydraulic model as one might be expecting. This is because the hazard modelling is done as a baseline while the data collected by the citizens/expert volunteers will be in near-real time so this will affect the early warning system, the involvement and hence awareness of citizens, and the updating of the hazard and risk information, which are all part of social vulnerability. We have added some additional text to section 3.1.2 on flood hazard modelling. We have also added an additional paragraph to the discussion regarding the contribution to citizens to the social vulnerability component to make this clearer.

One of the aspects that contributes most to a reduction in costs is related to the **emergency services.** Although the people involved in the emergency services are the same, they are employed in a much more efficient way as a result of the technology developed within the CO, which allows for better management of the teams responding to the event and the assignment of tasks based on an operator's location. For example, the authorities, in response to real-time reports from citizens and based on reliable model forecasts, can assign critical tasks to rescue teams, monitor the movements of the teams and assign new tasks to teams once a job is finished based on the proximity job principle. Secondly, among the factors of social vulnerability, the involvement of citizens contributes to changing those behaviours that are the

main causes of death and/or serious economic damage; an example would be actions such as trying to save your car during a flood event or trying to rescue your belongings from a flooded basement, which would be stopped or reduced as a result of the CO. We have added more text in the discussion to clarify how the CO improves the emergency services.

**Specific comments:**
**Overall there are too many abbreviations (e.g. L 214, 258, table 6, L 369, L 379). I was not able to find the definition of the abbreviation "EWS" (table 6).**

Response: Regarding L214, this is a blank line in our version of the paper unless you are referring to L215, which defines one of the variables in equation 2? Regarding L258, is the reviewer referring to FHR or Vp as an acronym? If it is FHR, this was defined further up in the text and used in equation 3. If Vp, then we have now added this to the caption of Figure 4. In Table 6, we have changed the acronym EWS to read Early Warning System as this was not defined previously in the paper so thank you for pointing this out. On L369, CLC refers to Corine Land Cover but we have now relaced the acronym with the words wherever they appear in the text. L379 defines variables that are then used in equations 5 and 6 and further in the text.

**L 55-59: Not all of the cited literature actually refers to a CO and the description of at least some of the stated studies is not accurate.**

Response: We have checked the referenced literature and deleted those references that do not refer to COs (Etter et al., 2018; Mazzonleni et al., 2017; Butaert et al., 2014) but to other forms of citizen science. Moreover, we have attributed the statement regarding the link of COs with authorities and policy more specifically as follows:

*"Specifically, Wehn et al. (2015) found that the characteristic links of COs to authorities and policy do not automatically translate into higher levels of participation in flood risk management, nor that communication between stakeholders improves; rather, changes towards fundamentally more involved citizen roles with higher impact in flood risk management can take years to evolve."*

**L97-100: How often do these observations get made and how many were collected in total? It would be very informative to include a photograph of such a "staff gauge with a QR code".**

During the WeSenseIt project, more than two hundred people were recruited for practical activities and were trained to use the WeSenseIt technologies. The data collected took place during the evaluation exercises organized to test the technology and to collect feedback for further development and to make improvements. From an exercise that took place between 15-18 Nov 2019, number of reports collected by those trained in the technology was around 1,100. During these events, the response of the volunteers was enthusiastic as well as their participation in sending environmental reports and information. Examples of photographs with the staff gauge and QR code have been added as a new Figure 1 to the paper.

**L 104: Did the volunteers operate the physical sensors? Or was this done by someone else?**

Response: No, the physical sensors are operated by AAWA in collaboration with the Regional Department for Soil Protection, the Environmental Agency, the Civil Protection Agency and their related professionals. We have added this information to the text.

**L 197 / table 1: It is not clear to me which of these data inputs are derived from citizen scientists and which are implemented anyway. Please make this distinction clearer so that the added value is more obvious.**

Response: None of the data in Table 1 (version posted online) are derived from citizen scientists. We have now added the input from citizen scientists to the Flood Vulnerability component as follows in the final row of Table 1.

These data are used indirectly in the calculation of social vulnerability, i.e., the Early Warning System (EWS) component through improvements in the reliability, lead time and information content of the EWS (Figure S2 in the Supplementary Information) as well as two components of Adaptive Capacity (Hazard and risk information updating and Citizen involvement - Figure S3 in the Supplementary Information).

**L 425: It would be helpful to add a range to this value, so as to show the associated uncertainty.**

Although we understand the point of the reviewer, we do not have a range around the expected average annual damage because it is already based on three probabilistic flood scenarios, so this damage value is an average across these scenarios. We recognize that there are multiple assumptions and uncertainties in this methodology, but we have not quantified them as such. Since the exercise is currently a hypothetical one but it provided sufficient evidence for funding of the CO for five years, going back and doing a full uncertainty analysis would not bring any further value for the operational running of the CO. Instead, once the CO is operational, it will then be more interesting for us to verify the results from the cost-benefit analysis. We do mention this future verification at the end of the Discussion and Conclusions section.

**L 430-432: Why do you think there is a difference, i.e. why is R3 and R4 reduced, but R1 and R2 increased? Add this to the discussion.**

Response: This occurs because the total area affected by the flood hazard is the same before and after implementation of a CO. What changes is the distribution between risk classes, i.e., R3 and R4 are reduced, which means that the areas at risk in classes R1 and R2 will increase. We added this to the results because it explains this finding directly after it is presented.

**Also table 9 does not show any areas, just damage, so the reference here probably refers to table 8?**

Response: Thank you for pointing out this error. We have corrected this but in the process of responding to reviewer 1, we have also combined tables, i.e., Table 8 and 10 are now combined (risk before and after the implementation of a CO in terms of area) and Table 9 and 11 (damage in euro amounts before and after the implementation of a CO). The text has also been updated accordingly.

**L 469: You mention that this method can be transferred to different catchments. It would be interesting to read your thoughts on what type of catchments this would be suitable for, e.g. what catchment scale.**

Once activated in the Brenta-Bacchiglione, the CO will also be extended to the other basins of the hydrographic district of the Eastern Alps, which are similar in size and hydrological characteristics. These are complex hydrographic basins with very variable regimes, from rapid response/torrential rainfall events of the alpine territories to the alluvial plain, which is composed of mountain and lowland river networks, artificial networks of reclamation and natural and/or artificial reservoirs. In general, we suggest

applying this methodology to catchments larger than 100 km$^2$. For catchments of this size and greater, we would have basin compatible response times, and hence, it would make sense to use model forecasts. We have added some of this text to the end of the discussion.

Another point to note is that for the application of the CO methodology, it is necessary that the population residing in the basin can be easily reached through such an initiative, and that they are familiar with, and are able to access, the technology (i.e., via a tablet, PC, smartphone).

**Technical corrections:**
**L 92: 7th (th in superscript)**

Response: Thank you for spotting this error. We have now corrected this.

[revised manuscript text omitted]

---

## Author Response (AR2)

**Minor Changes made to the Paper at the Request of One Reviewer/Editor**

Dear Editor,

The following minor changes have been made as requested:

- Table 3: It is not clear from the text how exposure values for which ranges are presented are handled in the analysis.

Response: we have added text to explain this – see page 8 of the manuscript with the text highlighted in yellow.

- L. 484-485: As discussed in the response to reviewers, you added that you will look for further funding after the initial project phase. In table S6 it is more or less clear that the costs are for a 5-year period. I think it would be good to repeat this in this sentence, so it is clear for the reader when the project will be evaluated and funding will be sought for continuation.
I think they just mean to add a sentence before Table S6 (because we already say this in the main text).

Response: we have added text to the Supplementary Material file – see page 9 of this file with the text highlighted in yellow.

Best wishes,

The authors

[revised manuscript text omitted]

| Category | $E_E$ | Description |
|---|---|---|
| Residential | 1 | The costs of restoring the buildings, those of the assets kept in them, and those relating to the accommodation of people during the restoration period are high. |
| Industrial | 1 | The costs of restoring buildings, those of the equipment present, and those of non-production during the period of inactivity are high. |
| Specialized agricultural | 0.3 - 1 | Average costs for the restoration of crops and the average costs for the lack of harvest. Not being able to differentiate between different types of crops, it was still necessary to differentiate between crops of very low value (e.g., corn: the harvest has a very low value and recovery times are short) and others of high value (e.g., vineyards: the harvest can be very valuable and recovery times can be very long). |
| Woods, meadows, pastures, cemeteries, urban parks | 0.3 | Low restoration costs and low costs related to the lack of timber production |
| Unproductive | 0.1 | Very low economic value |
| Ski areas | 0.3 | The costs of restoring the infrastructure and those related to the failure to use the service during the restoration period are low |
| Roads of primary importance | 1 | The costs of restoring the infrastructure and costs related to the failure to use the service during the restoration period are high |
| Landfill, Waste treatment plants, Mining areas, Purifiers | 0.5 | Average costs for the restoration of the various types of works and low costs regarding failure to carry out the service (e.g., in the event of a disaster, the wastewater ends up in the river and the waste is disposed of elsewhere) |
| Areas of historical, cultural and archaeological importance | 1 | High costs associated with the restoration of buildings |

Table S4: **The logic used to assign relative values to the categories of $E_{ECH}$**

| Category | $E_{ECH}$ | Description |
|---|---|---|
| Residential | 1 | High historical, cultural and environmental value |
| Industrial | 0.3 - 1 | Medium-high environmental value since the presence of industry is negative for the environment and therefore its disappearance would improve the environmental quality, but its damage could be negative as it could generate pollution. |
| Specialized agricultural | 0.7 | Medium environmental and cultural value |
| Woods, meadows, pastures, cemeteries, urban parks | 0.7 | Medium environmental and cultural value |
| Unproductive | 0.3 | If the unproductive land is affected by a hydrogeological event, the quality of the environment does not change as the environment is not substantially changed |
| Ski areas | 0.3 | Medium environmental value since the presence of the ski area is, in itself, negative for the environment and therefore its disappearance would improve the environmental quality, but it is instead positive from a recreational point of view |
| Roads of primary importance | 0.2 | The presence of primary roads is in itself very negative from a landscape point of view and therefore its disappearance would improve the environmental quality |
| Landfill, Waste treatment plants, Mining areas, Purifiers | 1 | High environmental value due to the pollution that would be generated in the event of service interruption and therefore a serious deterioration in environmental quality |
| Areas on which plants are installed as per Annex I of Legislative Decree 18 February 2005, n. 59 | 1 | High environmental value since the damage could be negative as it could generate pollution |
| Areas of historical, cultural and archaeological importance | 1 | High historical, cultural and environmental value |
| Environmental goods | 1 | High historical, cultural and environmental value |

**Table S5: Maximum flood damage values (€ / m²) per damage category (Huizinga, 2007)**

| Region/country | Residential building | Commerce | Industry | Road | Agriculture |
|---|---|---|---|---|---|
| EU27 | 575 | 476 | 409 | 18 | 0.59 |
| Italy | 618 | 511 | 440 | 20 | 0.63 |
| Luxembourg | 1443 | 1195 | 1028 | 46 | 1.28 |
| Germany | 666 | 551 | 474 | 21 | 0.68 |
| Netherlands | 747 | 619 | 532 | 24 | 0.77 |
| France | 646 | 535 | 460 | 21 | 0.66 |
| Bulgaria | 191 | 158 | 136 | 6 | 0.20 |

The Citizen Observatory (CO) for Flood Risk Management has an estimated cost of around 5 million Euros for a five-year period. After the five years, the CO will be evaluated, and further funding will be sought for the maintenance of the environmental monitoring network and IT platform and for the continuation of education and training campaigns. Table S6 provides a breakdown of these costs.

[revised manuscript text omitted]